# Beta oscillations and waves in motor cortex can be accounted for by the interplay of spatially structured connectivity and fluctuating inputs

**Ling Kang[1,2], Jonas Ranft[3], Vincent Hakim[1]\***

[1]Laboratoire de Physique de l'Ecole Normale Supérieure, CNRS, Ecole Normale Supérieure, PSL University, Sorbonne Université, Université de Paris, Paris, France; [2]School of Physics and Electronic Science, East China Normal University, Shanghai, China; [3]Institut de Biologie de l'Ecole Normale Supérieure (IBENS), CNRS, Ecole Normale Supérieure, PSL University, Paris, France

**\*For correspondence:**
vincent.hakim@ens.fr

**Competing interest:** The authors declare that no competing interests exist.

**Abstract** The beta rhythm (13–30 Hz) is a prominent brain rhythm. Recordings in primates during instructed-delay reaching tasks have shown that different types of traveling waves of oscillatory activity are associated with episodes of beta oscillations in motor cortex during movement preparation. We propose here a simple model of motor cortex based on local excitatory-inhibitory neuronal populations coupled by long-range excitation, where additionally inputs to the motor cortex from other neural structures are represented by stochastic inputs on the different model populations. We show that the model accurately reproduces the statistics of recording data when these external inputs are correlated on a short time scale (25 ms) and have two different components, one that targets the motor cortex locally and another one that targets it in a global and synchronized way. The model reproduces the distribution of beta burst durations, the proportion of the different observed wave types, and wave speeds, which we show not to be linked to axonal propagation speed. When the long-range connectivity or the local input targets are anisotropic, traveling waves are found to preferentially propagate along the axis where connectivity decays the fastest. Different from previously proposed mechanistic explanations, the model suggests that traveling waves in motor cortex are the reflection of the dephasing by external inputs, putatively of thalamic origin, of an oscillatory activity that would otherwise be spatially synchronized by recurrent connectivity.

## Editor's evaluation

This article makes a valuable contribution to the field. Here the authors have developed a convincing model to characterize the generation of motor cortex β oscillations. Using the model, the authors are able to recapitulate several properties observed experimentally. Given the long-standing interest in motor cortical β oscillations and current interest in traveling waves, this article will be of significant interest to the neuroscience community.

## Introduction

Neural rhythms are one of the most obvious feature of neural dynamics (*Buzsáki, 2006*). They have been recorded for more than 90 years in human (*Berger, 1929*) and they are a daily tool for the diagnosis of neurological dysfunction. Classic studies have shown that neural rhythms depend on neural structures and the behavioral state of the animal (*Adrian, 1935*). Examples include the theta rhythm,

the gamma rhythm in the visual cortex, or the fast 160–200 Hz cerebellar rhythm. However, in spite of their ubiquity and common diagnostic use, neural rhythms remain a somewhat mysterious feature of the brain dynamics. It remains to better understand how they are created and what they are a reflection of (*Wang, 2010*).

The beta rhythm consists of oscillations in the 13–30 Hz range. The motor cortex was found to be one of its main locations in early recordings in human subjects (*Jasper and Penfield, 1949*; *Penfield, 1954*). It was recorded in cats during motionless focused attention, when fixating a mouse behind a transparent screen (*Bouyer et al., 1987*). Subsequent studies in monkeys trained to perform an instructed-delay-task (*Murthy and Fetz, 1992*; *Sanes and Donoghue, 1993*) showed that beta oscillations develop in the motor and premotor cortex during the movement preparatory period and wane during the movement itself, in agreement with earlier observations in human (*Jasper and Penfield, 1949*; *Penfield, 1954*). It was also noted that beta oscillations did not have a constant amplitude in time but rather appeared as synchronized bursts of a few cycles of oscillations, often synchronized on electrodes 1–2 mm apart. More recently, it was observed using multi-electrode arrays that beta oscillations can come as planar (*Rubino et al., 2006*) or more complex (*Rule et al., 2018*; *Denker et al., 2018*) waves propagating horizontally on the motor cortex. Our aim in this paper is to develop a mechanistic framework for these observations and to compare it to available recordings in monkeys performing an instructed delayed reach-to-grasp task (*Denker et al., 2018*; *Brochier et al., 2018*). The model is based on recurrent interaction between local populations of excitatory (E) and inhibitory (I) populations of neurons coupled by longer range excitation. The local E-I module is well-known to exhibit oscillatory dynamics when the recurrent interaction within the E population is sufficiently strong. Long-range excitation between these local modules has been shown in previous works to synchronize the oscillatory dynamics of local E-I modules (*Ermentrout and Kopell, 1991*; *Kulkarni et al., 2020*). Comparison with recordings leads us to propose that the whole network is close to the oscillation threshold and that fluctuating inputs from other cortical areas or other structures, such as the thalamus, power the burst of beta oscillations. We show that such a spatially extended network submitted to both local and global external inputs exhibits waves of different types that closely resemble those recorded in monkey motor cortex. Our analysis makes testable, specific predictions on the structure of external inputs. More generally, it highlights the dynamical interplay between fluctuating inputs and intrinsic dynamics shaped by spatially structured connectivity, which is worthy of further experimental investigation.

## Data and modelling assumptions

We ground our modeling by comparing it to recordings obtained during a delayed reach-to-grasp task in two macaque monkeys (*Denker et al., 2018*) that have been made publicly available (*Brochier et al., 2018*). As in other recordings in similar tasks (*Murthy and Fetz, 1992*; *Sanes and Donoghue, 1993*; *Rubino et al., 2006*), beta oscillations are prominent during the 1 s waiting time, the movement preparatory period, and wane during the movement itself, as shown in *Figure 1—figure supplement 1a and c*. We thus focus on the recordings during the preparatory period in the following.

In order to develop a model of beta oscillations, assumptions have to be made on the source and mechanism of their generation. We explicitly list below our main ones and their rationale, since they differ from those made in some previous works.

Debate exists regarding the origin of beta oscillations in cortex. On the one hand, the basal ganglia display prominent beta oscillations during different phases of movement (*Leventhal et al., 2012*). Therefore, one view is that beta oscillations in the motor cortex are simply conveyed to the motor cortex from other structures such as the basal ganglia. On the other hand, sources of beta oscillations have been identified in the cortex (*Jensen et al., 2005*; *Sherman et al., 2016*). That different cortical regions have intrinsic oscillation frequencies matching those of their prominent rhythms is also supported by transcranial magnetic stimulation (TMS) perturbation studies. Specifically, the premotor area/supplementary motor area 6 has been found to resonate at ~30 Hz after stimulation by a short TMS pulse (*Rosanova et al., 2009*). This seems difficult to explain if the rhythm frequency originates from a distant structure. We thus adopt the intermediate view, previously advocated by *Sherman et al., 2016*, that beta oscillations are generated by recurrent interactions in the motor cortex but are strongly modulated by inputs from other cortical areas or other structures, notably thalamic ones.

Assuming that beta oscillations originate within M1, the question arises of the main neuronal populations and the recurrent interactions underlying rhythm generation. Two natural candidates are recurrent interactions between interneurons, or recurrent interactions between excitatory and inhibitory populations. Here, one important observation is that during the preparatory period of movement when beta oscillations are prominent, neurons do not fire periodically. As shown in *Figure 1—figure supplement 1b*, isolated units from the recordings of *Brochier et al., 2018* display large CVs. Beta oscillations thus appear to be a collective phenomenon arising from sparse synchronization (*Brunel and Hakim, 2008*) of different non-oscillating units. In such a regime, oscillations can emerge from recurrent interaction between interneurons, when inhibition is sufficiently strong. However, their frequencies are mainly controlled by the kinetics of synaptic transmission and tend to be in the upper gamma range or higher (*Brunel and Wang, 2003*; *Geisler et al., 2005*; *de Solages et al., 2008*). Lower frequencies arise in networks of E-I units, in which each of the two populations inhibits itself via a slower disynaptic loop. This leads us to assume that beta oscillations are sparsely synchronized oscillations arising from recurrent interaction between excitatory and inhibitory populations. Our aim is to account for recordings obtained from 4 mm × 4 mm multi-electrode arrays with 10 × 10 electrodes. Therefore, the spatial structure of the connectivity needs to be taken into account. We assume that each electrode records the activity of a local neuronal population. For computational tractability, we describe this population activity in a classic way by its firing rate (*Wilson and Cowan, 1972*; *Dayan and Abbott, 2005*), but we choose a particular firing rate formulation (see 'Methods') that was shown to agree well with direct simulations of spiking network models in previous works (*Kulkarni et al., 2020*; *Ostojic and Brunel, 2011*; *Augustin et al., 2017*). We assume that neurons under different electrodes are mainly connected by excitatory connections and that the connection probability decreases with distance (*Huntley and Jones, 1991*; *Capaday et al., 2009*; *Hao et al., 2016*).

## Results

### E-I modules connected by long-range excitatory connections exhibit oscillatory activity at beta frequency

The previous considerations lead us to study the model schematically depicted in *Figure 1a*, which generalizes one previously analyzed (*Kulkarni et al., 2020*). It consists of multiple recurrently coupled modules of excitatory (E) and inhibitory (I) neuron populations. These local modules are further coupled by long-range excitatory connections. The different modules are placed on a square 24 × 24 grid with the central 10 × 10 ones corresponding to the recording electrodes (see *Figure 1—figure supplement 2*). The decrease in synaptic connection probability with distance is described by the function $C(\mathbf{x})$, chosen in the numerical simulations to be Gaussian with range $l \sim 1$ mm, based on anatomical data and microstimulation studies (*Huntley and Jones, 1991*; *Capaday et al., 2009*; *Hao et al., 2016*). Notably, *Capaday et al., 2009* report a uniform distribution of boutons along horizontal axons, not a patchy one. A relatively slow propagation speed (~30 cm/s) along unmyelinated horizontal axons in the monkey cortex has been reported in previous works (*González-Burgos et al., 2000*; *Girard et al., 2001*). A comparable speed has been found for horizontal connections in the rat visual cortex (*Lohmann and Rörig, 1994*). We have taken this into account by introducing a delay proportional to distance, $D|\mathbf{x} - \mathbf{y}|$, between the activity $r_E$ in excitatory population at location ($\mathbf{y}$) and the corresponding postsynaptic depolarization in the E and I population at location $\mathbf{x}$. However, since data are predominantly available for the visual cortex and could be different for the motor cortex, we also compare the results in the following with the case of fast propagation and negligible propagation delays. The model is further detailed in 'Methods.' We begin by examining the dynamics of this network when external inputs are constant and the number of neurons in each module is very large. The effects of time-varying inputs and fluctuations due to the finite number of neurons in each module are then addressed in the following sections.

The network of E-I modules, described by *Equations 8–17*, has different dynamical regimes as a function of the synaptic weights $w_{EE}, w_{IE}, w_{EI}, w_{II}$ between the excitatory and inhibitory neural populations. This has been extensively studied in the rate model framework (*Ermentrout and Terman, 2010*; *Ashwin et al., 2016*) since the pioneering work of *Wilson and Cowan, 1972*. In the particular case of the present model ('Methods,' see also *Figure 1—figure supplement 3* for the f-I curve and adaptive time scale used), the different regimes are depicted in *Figure 1b*, generalizing previous

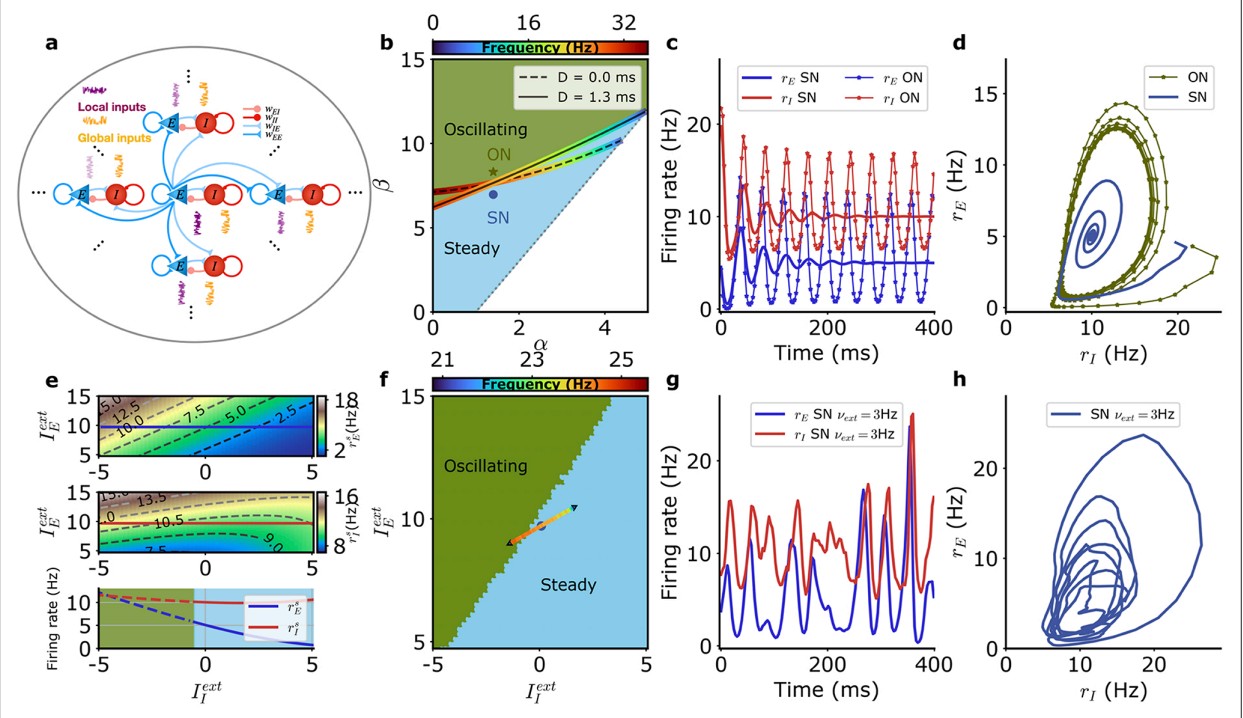

**Figure 1.** Model of neural network generating beta oscillations. (**a**) Schematic depiction of the model with excitatory populations (blue), inhibitory populations (red), independent external inputs on each module ('local,' purple), and inputs common to all modules ('global,' orange). (**b**) Different dynamical regimes for fixed firing rates of the excitatory and inhibitory populations ($r_E^s = 5$ Hz, $r_I^s = 10$ Hz), as a function of the strengths of recurrent excitation ($\alpha$) and of feedback inhibition through the E-I loop ($\beta$) for a fixed value of recurrent inhibition in the inhibitory population $\gamma = 2$, with the parameters $\alpha, \beta$ and $\gamma$ as defined in **Equation 34**. The oscillatory instability line for $D = 1.3$ ms (solid black) and $D = 0$ ms (dashed black) and the line of 'real instability' (short-dashed black) are shown. Color around the oscillatory lines indicates the frequency of oscillation along the bifurcation line. (**c**) Time series of the firing rate for the E (blue) and I (red) module populations at the SN (thick lines) and at the ON (thin lines with symbols) points. E and I population activities become steady at the SN point and display regular oscillations at the ON point. (**d**) Same data as in (**c**) for SN (blue) and ON (green) parameters but with $r_E$ plotted as a function of $r_I$. (**e**) Firing rates of the E (top) and I (middle) populations for SN parameters when the external inputs are varied and (bottom) along the solid lines in top and middle panels when only the external input on the inhibitory population $I_I^{ext}$ is varied (E blue, I red). The dashed parts of the line correspond to unstable steady states. (**f**) Different dynamical regimes as a function of the mean strength of external inputs for SN. The external inputs on each population vary along the colored line when the strength of the external input fluctuates. The color indicates the frequency associated with the linear dynamics at the respective stationary states. (**g**) Example of E and I activity time traces when the external inputs vary in time along the colored line in (**f**). (**h**) Same data as in (**g**) but with $r_E$ plotted as a function of $r_I$. Model parameters correspond to SN and ON in **Table 1**.

The online version of this article includes the following figure supplement(s) for figure 1:

**Figure supplement 1.** Recording data: LFP and single-unit characteristics.

**Figure supplement 2.** Numerical simulation grid.

**Figure supplement 3.** Rate model f-I curve and adaptive time scale.

**Figure supplement 4.** Dependence of the bifurcation lines and frequency on model parameters.

results (**Kulkarni et al., 2020**) to take into account the finite kinetics of synaptic current and propagation delays. Essential parameters controlling stability are the strength of recurrent excitation, the strength of feedback inhibition through the disynaptic E-I loop, and the strength of autoinhibition of interneurons on themselves, as respectively measured by parameters $\alpha, \beta$, and $\gamma$ (**Equation 34**). With other parameters fixed, a steady firing rate state in which excitatory and inhibitory populations fire at $r_E^s$ and $r_I^s$ is unstable when the strength of recurrent disynaptic self-inhibition $\beta$ becomes larger than a threshold value. Mathematical analysis (see 'Methods') provides the bifurcation line that separates the steady state regimes from the oscillatory ones. Its exact position depends on propagation delays as depicted in **Figure 1b** for zero delay and for the propagation delay, here chosen, of 1.3 ms between neighboring modules, see also 'Methods' **Equation 35**. The frequency of oscillations that arise when crossing the bifurcation line at a particular point is also shown in **Figure 1b**. When recurrent excitation

**Table 1.** Parameter table.

**Parameters**

| Symbol | Value | | | | Unit | Definition |
|---|---|---|---|---|---|---|
| | SN | SN' | ON | $SN_0$ | | |
| $r_E^s, r_I^s$ | 5, 10 | | | | Hz | Steady firing rates |
| $I_E^s, I_I^s$ | −6.28, −3.62 | | | | mV | Currents at $r_E^s, r_I^s$ |
| $\tau_E^s, \tau_I^s$ | 8.74, 7.14 | | | | ms | Adaptive membrane time constant at $r_E^s, r_I^s$ |
| $\Phi'_E(I_E^s), \Phi'_I(I_I^s)$ | 1.46, 2.30 | | | | Hz/mV | Firing rate gains at $r_E^s, r_I^s$ |
| $\tau_r^E, \tau_r^I$ | 0.70 | | | | ms | Rise time of synaptic currents |
| $\tau_d^E, \tau_d^I$ | 3.50 | | | | ms | Decay time of synaptic currents |
| $\tau_l^E, \tau_l^I$ | 0.50 | | | | ms | Latency of synaptic currents |
| $l$ | 2 | | | | | Excitatory connectivity range |
| $N_E, N_I$ | 16000, 4000 | | | | | Neuron numbers in each E-I module |
| $\tau_{ext}$ | 25 | | | | ms | Correlation time of external input fluctuations |
| $\nu^{ext}$ | 3 | | | | Hz | External input amplitude fluctuations |
| $I_E^{ext,0}, I_I^{ext,0}$ | 9.72, 0.08 | 6.12, 0.08 | 13.72, 0.08 | 5.72, 0.08 | mV | External currents |
| $w_E^{ext}$ | 0.96 | 1.12 | 0.96 | 1.20 | mV·s | External input onto excitatory neurons synaptic coupling strength |
| $w_I^{ext}$ | 2 | 2 | 2 | 2 | mV·s | External input onto inhibitory neurons synaptic coupling strength |
| $w_{II}$ | 0.87 | 0.87 | 0.87 | 0.87 | mV·s | Recurrent synaptic coupling strength (I to I) |
| $w_{IE}$ | 1 | 1 | 1 | 1 | mV·s | Recurrent synaptic coupling strength (E to I) |
| $w_{EE}$ | 0.96 | 1.12 | 0.96 | 1.20 | mV·s | Recurrent synaptic coupling strength (E to E) |
| $w_{EI}$ | 2.08 | 1.80 | 2.48 | 1.80 | mV·s | Recurrent synaptic coupling strength (I to E) |
| $D$ | 1.30 | 1.30 | 1.30 | 0 | ms | Propagation delay between to nearest E-I modules |
| $c$ | 0.40 | 0.30 | 0.40 | 0.30 | | Proportion of global external inputs |

increases, the oscillation frequency decreases. As seen in **Figure 1b**, propagation delays displace the bifurcation lines but do not change much how the oscillation frequency depends on recurrent excitation. The positions of the bifurcation lines in the $(\alpha, \beta)$ parameter space also depend on the strength $\gamma$ of recurrent inhibition between interneurons as shown in **Figure 1—figure supplement 4a**. The oscillation frequency on the bifurcation line also increases as $\gamma$ becomes larger (**Figure 1—figure supplement 4**). The latency, rise time, and especially the decay time of the synaptic currents are other parameters that influence the oscillatory bifurcation as shown in **Figure 1—figure supplement 4d–k**. Quite generally, the oscillation frequency decreases when the synaptic time constants increase.

For the parameters of *Figure 1* (see *Table 1*), the dynamics of the network is illustrated for two values of recurrent inhibition, one above (parameters ON, 'oscillating network') and one below (parameters SN, 'steady network') the bifurcation line point corresponding to beta oscillation frequency (~20 Hz). As illustrated in *Figure 1c and d*, for recurrent inhibition stronger than the critical value, the ON model oscillates regularly. For a lower value of recurrent inhibition (SN model), the firing rates of the excitatory and inhibitory populations steadily fire at constant rates. When perturbed away from these rates, the firing rates relax to their stationary values in an oscillatory fashion, transiently exhibiting beta oscillations.

For fixed synaptic coupling, the steady-state firing rates of the excitatory and inhibitory neuronal populations depend on the strengths of the external inputs on those two populations. This is quantitatively illustrated in *Figure 1e* for synaptic parameters of model SN. Interestingly, when the external input on the inhibitory neurons is increased, their steady firing rate and the steady firing rate of the excitatory population both decrease for a large range of the external input current. (The firing rate of the inhibitory population starts to increase again for very large inputs, when the firing rate of the excitatory population tends to vanish.) This 'paradoxical suppression of inhibition' (*Tsodyks et al., 1997*) is typical of inhibition-stabilized networks and is observed in many cortical areas (*Ozeki et al., 2009*; *Sanzeni et al., 2020*) including motor cortex. It arises in the SN model because it is only when recurrent excitation is large, as measured by the parameter α, that its frequency of oscillation is in the beta range (*Figure 1b*). Namely all networks with $\alpha > 1$ cannot stably fire at moderate rates without being stabilized by inhibition.

When the external input strengths are changed on the excitatory and inhibitory populations, the steady discharge rates of the two populations change accordingly. They can enter into the parameter regime where a steady discharge is actually unstable and is replaced by an oscillatory one, as shown in *Figure 1f*. We consider in the present model excitatory inputs to the motor cortex coming from a distal area. The respective strengths of the external inputs, $I_E^{ext}$ and $I_I^{ext}$, on the excitatory and inhibitory populations are proportional to the respective strengths of their synapses, $w_E^{ext}$ and $w_I^{ext}$. Thus, $I_E^{ext}$ and $I_I^{ext}$ vary with the activity of the distal area on a line in the $(I_E^{ext}, I_I^{ext})$ plane as depicted in *Figure 1f*. The respective strengths of $w_E^{ext}$ and $w_I^{ext}$ are taken in the model (*Equation 37*) such that the network enters the oscillatory regime when the external inputs are decreased as shown in *Figure 1f*. This conforms to the early observation (*Jasper and Penfield, 1949*; *Penfield, 1954*) that external inputs suppress beta oscillations in motor cortex.

When the external input strength on the SN model varies in time, its operating point moves relative to the bifurcation line and correspondingly, beta oscillations are dampened at varying rates. This leads to waxing and waning of their amplitudes as shown in *Figure 1g and h*. This is also true for the ON model that stands close to the bifurcation line and can move into the non-oscillatory regime when external inputs vary. These networks that operate close to the bifurcation line thus appear as promising candidates to describe the waxing-and-waning of beta oscillations seen in recording data. This leads us to try and compare beta oscillations produced in the model by inputs varying on an appropriate time scale, together with intrinsic fluctuations arising from the finite number of neurons in each module, to those seen in electrophysiological recordings.

## Fluctuation of external inputs and finite-size stochasticity produces model LFP signals statistically similar to the recorded ones

Early reports (*Murthy and Fetz, 1992*) already stressed that beta oscillations were not of constant amplitude but modulated on different time scales. As confirmed in several later studies (see, e.g., *Feingold et al., 2015*), the average power of beta oscillations changes on a second time scale with different phases of movement, for example, preparation, execution, and post-execution periods. Within individual trials, the amplitude of beta oscillations fluctuates on much shorter time scales with brief bursts of elevated amplitudes of ~100 ms duration. Examples of LFP spectrograms from two trials of *Brochier et al., 2018* are shown in *Figure 2a and b* and *Figure 2—figure supplement 1a and b*, respectively, for monkeys L and N. Whereas single trials exhibit short bursts of beta oscillations, trial-averaged spectrograms (*Figure 2c* and *Figure 2—figure supplement 1c*) and LFP power spectra (*Figure 2d* and *Figure 2—figure supplement 1d*) only show the average increase of oscillation in the beta band and its modulation on a 1 s behavioral time scale (*Figure 2c* and *Figure 2—figure supplement 1c*). A more quantitative characterization of the brief high oscillation amplitude events, the beta

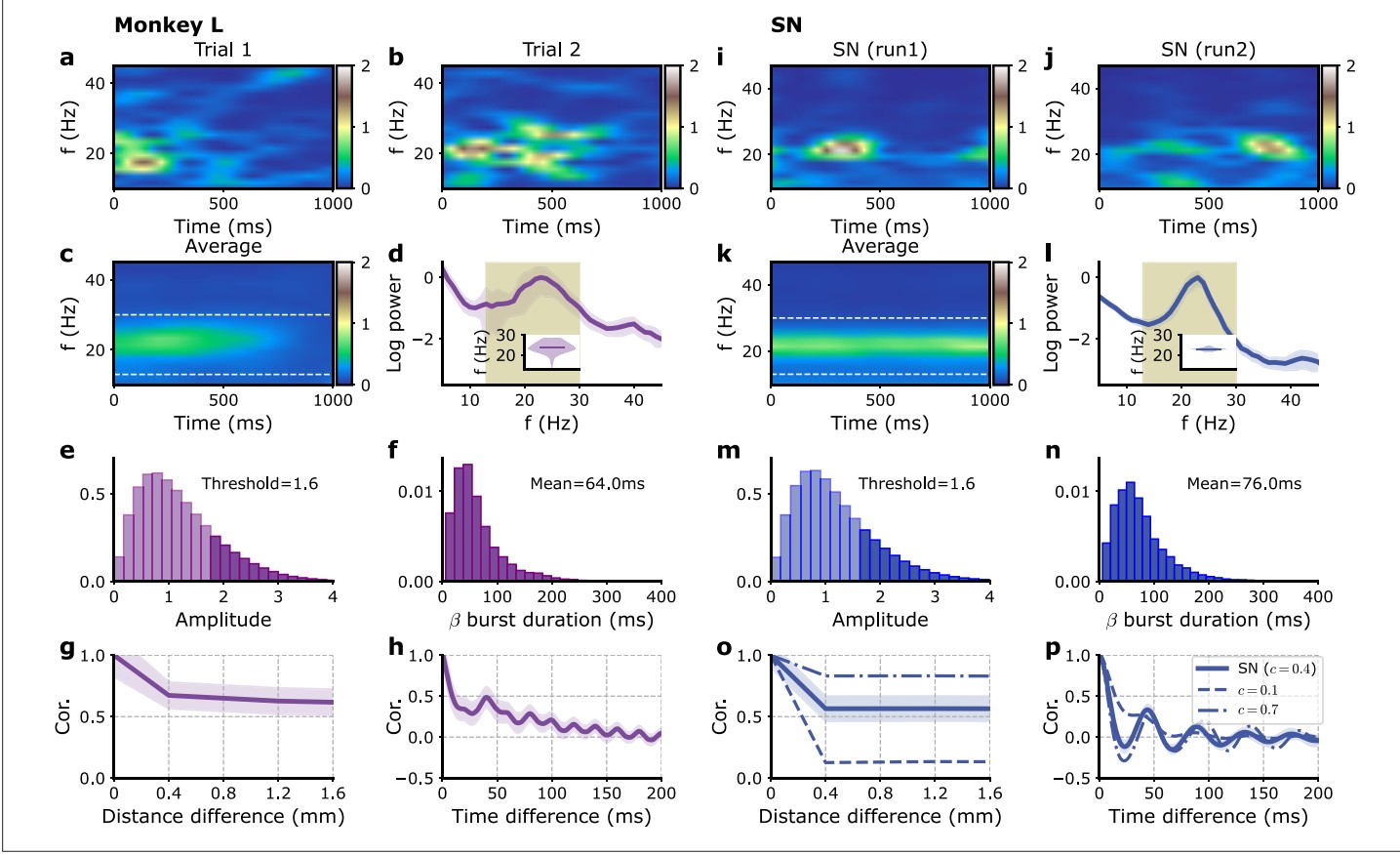

**Figure 2.** Single-electrode recordings vs. model E-I module dynamics with fluctuating inputs. (**a–p**) Monkey L recordings (**Brochier et al., 2018**), (**i–p**) model simulations. (**a, b**) Two examples of single trial spectrograms of a single-electrode LFP during the preparatory period (the CUE-OFF to GO period in **Figure 1—figure supplement 1**). (**c**) Spectrogram averaged over different trials and different electrodes. (**d**) Power spectrum of single-electrode LFP averaged over all electrodes. Average over trials (solid violet line) and standard deviation of fluctuation over trials (violet shaded region). Inset: violin plot showing the distribution over trials of the average power spectrum peak frequency in the 13–30 Hz interval. (**e**) Distribution of beta oscillation amplitudes. The amplitude of oscillation corresponding to the beta burst are shown in darker color. (**f**) Distribution of beta burst duration. (**g**) Average LFP cross-correlation between two electrodes as a function of their distance. (**h**) Auto-correlation of single LFP time-series averaged over trials and electrodes. (**i–p**) Corresponding panels for simulations with model SN. (**i, j**) Two examples of single modules spectrograms of $I_E$ 1 s time series. (**k**) Average $I_E$ time series spectrogram. (**l**) Corresponding power spectrum (see **d** for a detailed description). (**o**) Cross-correlations between different modules $I_E$ times series as a function of their distance, for different fractions $c$ of global (i.e., shared) external inputs, $c = 0.4$ (solid), $c = 0.1$ (dashed), and $c = 0.7$ (dashed-dotted). (**p**) Auto-correlation of single module $I_E$ time series for the different fractions $c$ in (**o**). Model parameters correspond to SN in **Table 1**.

The online version of this article includes the following figure supplement(s) for figure 2:

**Figure supplement 1.** Beta bursts and power spectra for monkey N.

**Figure supplement 2.** Data analysis: filtering and burst amplitude threshold.

**Figure supplement 3.** Model power spectra and beta bursts as a function of the external input parameters.

bursts, here taken to be the 75th percentile of the beta oscillation amplitude distribution (**Figure 2e** and **Figure 2—figure supplement 1e**, see also **Figure 2—figure supplement 2**), is provided by their duration distributions shown in **Figure 2f** and **Figure 2—figure supplement 1f**.

Can the model introduced in the previous section account for these data? At least two sources can be considered for these transient bursts of oscillations fluctuating from trial to trial. The first one is that the number of neurons in each E-I module, the population of neurons in a 200 μm neighborhood of each electrode, is finite, of about $2 \times 10^4$ cells given the cell density in cortex. This by itself produces stochastic fluctuations of each module's activity. These intrinsic fluctuations can be described in a simple way by adding to each neuronal population's mean firing rate a stochastic component that is inversely proportional to the square root of its population size (see 'Methods'; **Kulkarni et al.,**

*2020*; *Brunel and Hakim, 1999*). The effects of these intrinsic fluctuations for models ON and SN (*Figure 1b*) can be obtained both mathematically and through computer simulations. The results are displayed in *Figure 2—figure supplement 3* for varying numbers $N$ of neurons per module around our estimated value $N = 2 \times 10^4$. In this figure and following ones, we plot the excitatory current in the excitatory population as a simple proxy for the LFP in the model. We refer to it as the model 'proxy-LFP' in the following. For the parameters SN (*Figure 2—figure supplement 3a and b*), the model LFP power spectrum amplitude decreases with the number of neurons, but in the whole range $N = 2 \times 10^3 - 2 \times 10^5$, the power spectrum shape is invariant and accurately coincides with the one obtained mathematically by a linear analysis (*Equation 51*). Compared to the experimental LFP power spectra (*Figure 2d* and *Figure 2—figure supplement 1d*), the peak around beta frequencies is less pronounced; furthermore, the power spectrum misses the trough and enhancement for frequencies lower than beta frequency observed in the recordings. For the (oscillatory) ON model (*Figure 2—figure supplement 3j and k*), the power spectra are more strongly peaked than the experimental ones, with in addition a notable peak at the harmonic frequency of ~50 Hz coming from the non-sinusoidal shape of the oscillations (*Figure 1c and d*). This indicates that in either the SN or the ON model, intrinsic fluctuations are not sufficient to account for the experimental spectra of *Figure 2d* and *Figure 2—figure supplement 1d*.

Besides intrinsic dynamical fluctuations, it is likely that fluctuations in the motor cortex arise from time-varying inputs coming from other cortical areas or extra-cortical structures, or a mixture of the two. In absence of specific data, we model these as fluctuating inputs with a finite correlation time, more precisely as Ornstein–Uhlenbeck processes with a relaxation time $\tau_{ext}$ ('Methods') and refer to them simply as 'external inputs.' The network dynamics also depend on the way these external inputs target the motor cortex. Are they essentially independently targeting local areas or do they target the motor cortex in a more globally synchronized manner? Again in the absence of specific measurements, we suppose that the external inputs comprise a mixture of global inputs that provide a fraction $c$ of the power, and independent local inputs that provide the remaining fraction $1 - c$ (see 'Methods).

With this simple description, external inputs are characterized by three parameters: their amplitude $\nu_{ext}$, their correlation time $\tau_{ext}$, and the fraction $c$. We set out to determine these parameters by comparison with the single-electrode LFP trace and with the cross-correlation between the LFP traces of different electrodes.

The single-electrode power spectra and the single-electrode oscillation depend little on the fraction $c$, we thus consider them first (*Figure 2—figure supplement 3*).

The inclusion of external fluctuations increases the lower frequency part of the power spectra. It also comparatively decreases the power at frequencies higher than beta frequencies as soon as the fluctuation correlation time is in the few tens of millisecond range as shown for models SN (*Figure 2—figure supplement 3c–f*) and ON (*Figure 2—figure supplement 3l–o*). The amplitude of external fluctuations should be large enough for their effect to be of larger magnitude than the one produced by intrinsic fluctuations namely $\nu_{ext} > 0.2$ Hz. For the SN model, it should also be small enough ($\nu_{ext} < 3$ Hz), not to produce a secondary power enhancement at double the beta frequency as shown in *Figure 2—figure supplement 3c*. In this range of amplitudes, the power spectrum shape is independent of the external fluctuation amplitude $\nu_{ext}$ after normalization and corresponds to the mathematical expression (*Ozeki et al., 2009*; *Figure 2—figure supplement 3d*). For the ON model, a sharp secondary peak at twice the beta frequency is present at low external fluctuation amplitude (*Figure 2—figure supplement 3l*). When the external fluctuations are stronger, this peak is blurred and becomes a wide power enhancement around twice the beta frequency for $\nu_{ext} \sim 3$ Hz (*Figure 2—figure supplement 3f and o*), comparable to the one seen for the SN model at the same external fluctuation amplitude (*Figure 2—figure supplement 3c*). The external amplitude fluctuations also produce the power spectrum trough and enhancement observed at lower frequencies in experimental recordings (*Figure 2d* and *Figure 2—figure supplement 1d*) when their correlation time is not too short (*Figure 2—figure supplement 3e–f and n–o*).

Examination of the burst durations and amplitudes provides further information and constraints on the parameters. For both SN and ON models, the burst mean duration grows with the input correlation time $\tau_{ext}$ (*Figure 2—figure supplement 3g and p*) while it is weakly dependent on the input fluctuation amplitude $\nu_{ext}$ (*Figure 2—figure supplement 3i and r*). For the SN model, the input fluctuation amplitude $\nu_{ext}$ does not strongly affect the burst amplitude and duration distributions

(*Figure 2—figure supplement 3h–i*). Their shapes are moreover close to the experimentally observed ones (*Figure 2e and f*). On the contrary, for the ON model, the shape of the burst amplitude distribution strongly depends on the amplitude $\nu_{ext}$ (*Figure 2—figure supplement 3q*). It is only for large $\nu_{ext}$ that the shape is close to the one obtained for the SN model and resembles the experimental one. The coincidence of the SN and ON distributions, as for the power spectra, is indeed expected when the fluctuations are large enough as compared to the difference between their reference parameters. Since the ON network appears potentially relevant only for large fluctuations, when the distinction before the two network set points is not meaningful, we focus in the following on a network at the set point SN, that is fluctuating around a non-oscillatory set point.

*Figure 2i–n* and *Figure 2—figure supplement 1i–n* show the results of model simulations for the SN model with a relaxation time $\tau_{ext} = 25$ ms and $\nu_{ext} = 3$ Hz. Both LFP power spectra and beta burst duration as well as amplitude distributions closely resemble the experimental ones.

Having reproduced the single-electrode characteristics of the recording, we turn to the equal-time correlation between signals on different electrodes. As shown in *Figure 2g* and *Figure 2—figure supplement 1g*, the correlation between two neighboring electrodes is lower than the autocorrelation of a single-electrode LFP, namely, its variance, but is comparable to the correlation between more distant electrodes. Namely, part of the LFP signal is strongly synchronized between distant electrodes. This is not the case in the model if the external inputs on different modules are uncorrelated, that is, for $c = 0$. At the other extreme, when $c = 1$, the signals are much more correlated than in the data. As shown in *Figure 2o*, it is only when the external inputs are almost equally split between local and global contributions that the model correlations are comparable to the experimental ones. For monkey L (N), the value $c = 0.4$ ($c = 0.3$) is found to provide the best match. (Note that for the comparison with monkey N, we also use slightly changed synaptic couplings SN' that lead to a slightly lower peak beta oscillation frequency in accordance with experimental data, see *Table 1* for retained parameter values.) The autocorrelation of the single-electrode LFP (*Figure 2h* and *Figure 2—figure supplement 1h* for monkeys L and N, respectively) is also well accounted for by the model (*Figure 2p* and *Figure 2—figure supplement 1p* for parameters SN and SN', respectively).

*Figure 2* and *Figure 2—figure supplement 1* provide a comparison between the experimental data for the two monkeys and the model at points SN and SN', respectively, with the determined characteristics of the external inputs. The model clearly accounts well for the data characteristics, as quantified by the power spectra, burst duration, and amplitude distributions and correlation between different electrodes.

This leads us to investigate how the model compares to the data for other spatio-temporal characteristics of the LFP signals, namely, traveling waves of activity of different types and their characteristics.

## Traveling waves of different types

Several groups have reported traveling waves on the motor cortex during the preparatory phase of the movement in instructed-delay reaching tasks. In the early study of *Rubino et al., 2006*, planar waves of beta oscillations in multielectrode LFP recordings were described. Later works (*Rule et al., 2018*; *Denker et al., 2018*) have classified the spatio-temporal beta oscillations recorded by multielectrode arrays into different states. Here, we use the data of *Brochier et al., 2018* and closely follow the classification scheme proposed by these authors (*Denker et al., 2018*), distinguishing periods with planar (Pla.) or radial (Rad.) waves, as well as globally synchronized (Syn.) episodes and more random (Ran.) appearing states. The classification criteria for these four states are based on the instantaneous phase and phase gradient spatial distributions as precisely described in 'Methods.' In short, negligible phase delays between electrodes characterize the synchronized state. In the other states, the oscillatory activities of some electrodes have significant phase differences and give rise to spatial phase gradients. The phase gradients are tightly aligned for planar waves, pointing inward or outward for radial waves, or are more disordered in the remaining 'random' category. We apply the exact same analysis to the experimental recordings and our model simulation data (*Figure 3—figure supplement 1* and 'Methods').

*Figure 3a* provides one example of a planar wave in monkey L recordings (*Brochier et al., 2018*) with the narrow distribution of phase gradient directions shown in *Figure 3b*. An example of a radial wave is shown in *Figure 3c* with its outward pointing phase gradients shown in *Figure 3d*. Analogous examples for monkey N are shown in *Figure 3—figure supplement 2*.

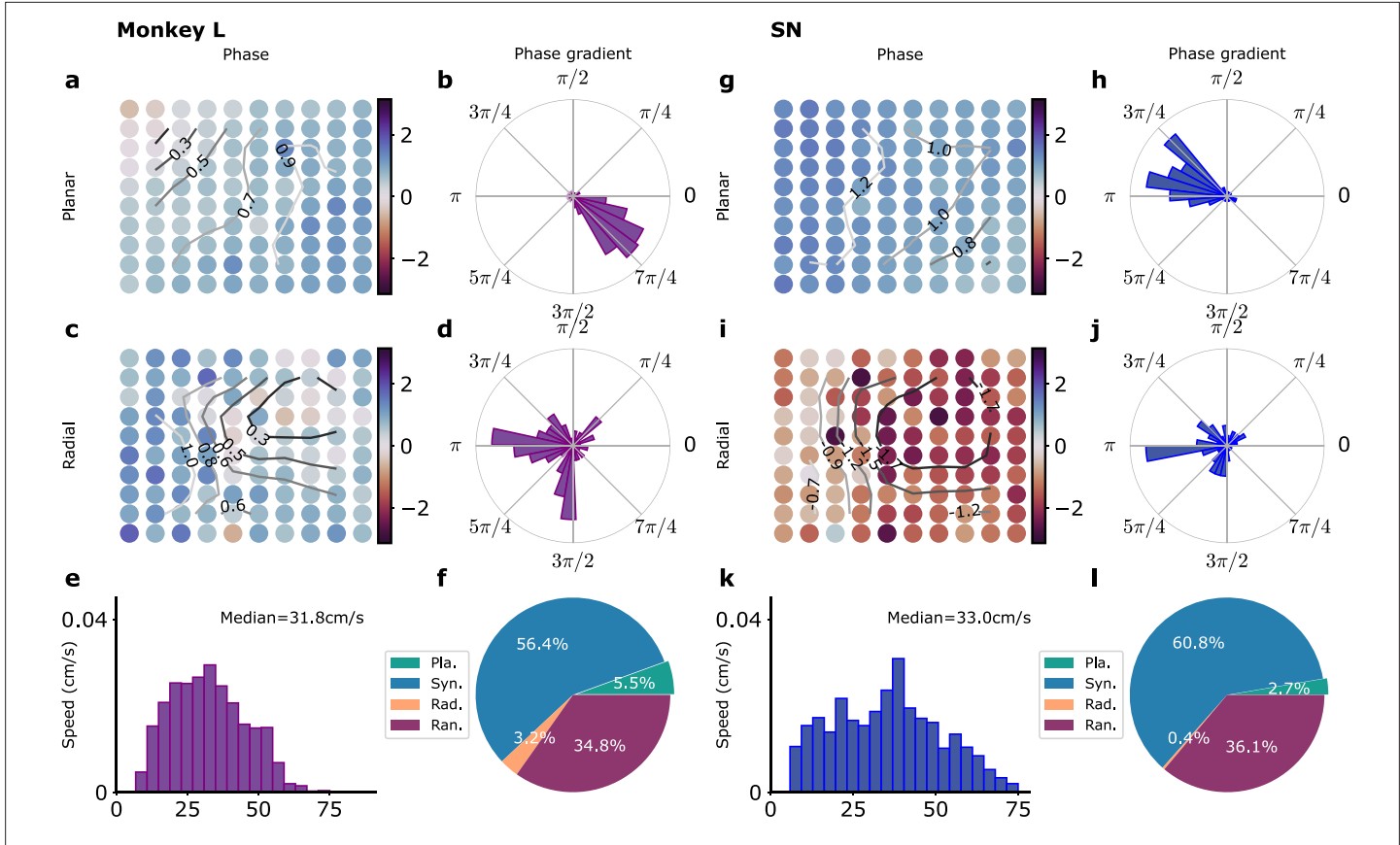

**Figure 3.** Waves in recordings and in model simulations. (**a–f**) Monkey L recordings (**Brochier et al., 2018**), (**g–l**) Model simulations with SN parameters. (**a**) Snapshot of a planar wave. The phases of the LFP on the different electrodes are shown in color. Smoothed phase isolines are also shown (thin lines). (**b**) Distribution of phase gradients on the multielectrode array in (**a**). Note the high coherence of the phase gradients ($\sigma_g = 0.61$). (**c**) Example of a radial wave with LFP phases and isolines displayed as in (**a**). (**d**) Corresponding distribution of phase gradients ($\sigma_g = 0.13$). (**e**) Distribution over time and trials of measured planar wave speeds. (**f**) Proportion over time and trials of planar (Pla.), synchronous (Syn.), radial (Rad.), and random (Ran.) wave types. (**g–l**) Corresponding model simulations. (**g**) Snapshot of a planar wave showing the $I_E$ phases of different modules. (**h**) Distribution of $I_E$ phase gradients ($\sigma_g = 0.51$) in (**g**). (**i**) Snapshot of a radial wave. (**j**) Distribution of phase gradients in (**i**) ($\sigma_g = 0.13$). (**k**) Distribution of planar wave speeds. (**l**) Proportions of different wave types.

The online version of this article includes the following figure supplement(s) for figure 3:

**Figure supplement 1.** Data analysis protocol.

**Figure supplement 2.** Waves in model simulations and in recordings for monkey N.

**Figure supplement 3.** Variation of wave type distribution and planar wave speed with the amplitude of the external input fluctuations ($\nu_{ext}$).

**Figure supplement 4.** Waves statistics in monkey L recordings.

**Figure supplement 5.** Waves statistics in SN model simulations.

**Figure supplement 6.** Waves statistics in monkey N recordings.

**Figure supplement 7.** Waves statistics in SN' model simulations.

**Figure supplement 8.** Simulation of the model of **Figure 3** with periodic boundary conditions.

**Figure supplement 9.** Intrinsic stochasticity without local inputs.

**Figure supplement 10.** Proportion of different waves for shuffled signals.

**Figure supplement 11.** Model with no propagation delay.

**Figure supplement 12.** Variation of the excitatory connectivity range, $l$, in the model of **Figure 3—figure supplement 11**.

For a traveling wave of oscillatory activity at frequency $f$, the local phase velocity is directly related to the inverse of the local phase gradient $\nabla\phi$. Following previous works (**Rubino et al., 2006**; **Rule et al., 2018**; **Denker et al., 2018**), we define the average wave speed in the presence of phase gradients varying from electrode to electrode as

$$v = \frac{2\pi f}{\sum_{x,y} |\nabla\phi(x,y)|/N} \tag{1}$$

where the denominator is the gradient magnitude averaged over all $N$ electrode positions. We refer to $v$ simply as the wave speed. The distribution of observed planar wave speeds is shown in **Figure 3e**. The median planar wave speed is about 30 cm/s but it should be noted that the distribution includes also events with much smaller velocities. The proportion of the different wave types is depicted in **Figure 3f**.

Can these data be accounted for by our model network? As discussed in the previous section, the single electrode LFP data and their correlations determine the correlation time ($\tau_{ext}$) of the fluctuating inputs and their repartition ($c$) between global and local inputs, but do not tightly constrain their amplitude. We thus performed a series of model simulations with different input amplitudes $\nu_{ext}$ for the SN model as summarized in **Figure 3—figure supplement 3a**. The four wave states can be observed for most $\nu_{ext}$. Examples of planar and radial waves are provided in **Figure 3g–j**. However, the proportion of the different wave types and the planar wave speeds greatly vary with $\nu_{ext}$ (**Figure 3—figure supplement 3**). In the SN model, the input fluctuations promote oscillations. The stronger the amplitude $\nu_{ext}$, the more developed the oscillations in neural activity and the better they synchronize. Thus, increasing the noise amplitude first increases the proportion of synchronized activity and the propagation speed of planar waves. At higher noise amplitudes, the desynchronizing effect of noise counteracts its oscillation-promoting effect and the wave speed decreases with increasing input amplitude. This leads in the SN model for $\nu_{ext}$ in the range $1-3$ Hz to a proportion of a planar wave states of a few percent with a planar wave speed of about 30 cm/s, as observed in the experimental recordings (**Figure 3e–f**). As a consequence, the proportion of planar and radial waves as well as synchronized activity increases with $\nu_{ext}$. The effect of increasing the amplitude of the input fluctuations is quite different for the ON model (**Figure 3—figure supplement 3b**). In this case, the oscillatory activity is self-generated and the oscillations between different modules are well-synchronized without fluctuating inputs. The local inputs tend to desynchronize the different modules and create oscillatory phase differences between them. The proportion of fully synchronized states thus decreases with increasing $\nu_{ext}$. The speed of planar waves that is inversely proportional to the magnitude of phase gradients (**Equation 1**) also decreases. For an input fluctuation amplitude $\nu_{ext} = 3$ Hz, the SN model agrees well with the distribution of observed wave speeds and the repartition between the different wave types for monkey L as shown in **Figure 3**, although radial waves seem somewhat underrepresented in the model network. This is also the case for monkey N as shown in **Figure 3—figure supplement 2**. The detailed statistics of the duration, wave speeds, oscillation amplitude, and frequency for the four wave states are detailed in **Figure 3—figure supplement 4** and **Figure 3—figure supplement 5** for monkey L recordings and the SN model and in **Figure 3—figure supplement 6** and **Figure 3—figure supplement 7** for monkey N recordings and the SN' model. The SN model simulated in **Figure 3** and **Figure 3—figure supplement 5** is taken as our reference model in the following and referred to as such. All its parameters are provided in **Table 1**.

The model and the simulation results suggest that the observed spatio-temporal patterns arise from the opposing effects of synchronization by long-range excitatory connectivity and global inputs and desynchronization by local inputs and the intrinsic stochasticity arising from the finite number of neurons in each module.

Before proceeding, it appears worth discussing and scrutinizing different aspects of our simulations. At a technical level, the simulation results come from the central $10 \times 10$ modules of a larger $24 \times 24$ grid with fixed boundary conditions (**Figure 1—figure supplement 2** and 'Methods'). In order to test the effect of the boundary conditions on the results, we performed a set of simulations and measurements with the same setting but with periodic boundary conditions. As shown in **Figure 3—figure supplement 8**, this did not significantly change the auto- and cross-correlations of module activities, nor the distributions of different wave types or of planar wave speeds. Therefore, one can conclude that the results are not dependent on the implemented boundary condition.

At a more conceptual level, one can wonder whether all the model components are necessary. A first natural question is whether intrinsic stochasticity would be enough to create spatio-temporal patterns similar to experiments without the necessity for desynchronization coming from local inputs. In order to address this possibility, we performed a set of simulations without local inputs. We increased the intrinsic noise as compared to our reference model ($N = 20000$) by lowering the numbers of neurons per module. As shown in *Figure 3—figure supplement 9*, for a number of neurons of $N = 2000$ (or higher) per module, the dynamics is too synchronized to account for the experimental data, the cross-correlation of the activity of different modules is very high even for distant modules and the speed of propagation of planar waves is also quite high. If the amplitude of the fluctuating global inputs to the network is decreased to ease the desynchronization of different modules, the network stays too close to the operating point SN and planar waves are no longer seen (not shown). Results comparable to our reference model can only be obtained when the number of neurons per module is lowered to $N = 200$ (*Figure 3—figure supplement 9a–d*). This of course would be quite a low number for the $0.1 - 0.2$ mm$^2$ area of motor cortex that each module is meant to represent, even if restricted to the cortical layers (e.g., II/III) where the dynamics could mainly take place. We found in a previous work (*Kulkarni et al., 2020*) that our reduced description of local noise (*Equation 10* and 'Methods') agreed well with simulations of spiking networks with dense local connectivity and sparse long-range connectivity. Whether our description of intrinsic stochasticity underestimates the fluctuations in a more sparsely connected network would have to be checked by large direct numerical simulations of spiking networks. Even if this was the case, the very low number of neurons in a module that we found to be necessary to reproduce the experimental recordings would still support the role of local inputs in creating dephasing between different local regions of motor cortex and planar waves.

A second legitimate question concerns the role of spatial correlations, produced by the connectivity spatial structure, in the observed spatio-temporal dynamics. Could the results be simply the product of random fluctuations in the activities of different modules? In order to address this question, we simulated the model of *Figure 3* and shuffled the proxy-LFP signals of the different modules after the simulation. We then reclassified the different types of waves in the shuffled signals. As a comparison, we followed the same procedure for the recordings of monkey L, namely, we classified the occurence of different types of waves after shuffling the LFPs recorded on different electrodes. The results are shown in *Figure 3—figure supplement 10* and very similar for the simulations and the experimental recordings. Shuffling has little effect on the proportion of synchronized and random waves. This is intuitively understandable since synchronized signals remain synchronized after shuffling, and shuffling cannot synchronize unsynchronized signals. More interestingly, shuffling entirely suppresses the appearance of planar waves. Therefore, the spatial correlation of different electrode signals, a reflection of the underlying long-range connectivity, is crucial for the appearance of planar waves. This conclusion is further supported by varying the connectivity range as described below.

## Properties of planar waves and characteristics of their inputs

The presence of planar waves is one remarkable feature of the multi-electrode recordings. It is thus worth analyzing some of their properties in the model.

The mean speeds of planar waves in the recordings and in the model are of a few tens of cm/s (*Figure 3e and k* and *Figure 3—figure supplement 2e and k*), which is comparable to the propagation speed along non-myelinated horizontal fibers (*González-Burgos et al., 2000*; *Girard et al., 2001*). One might thus think that both are tightly linked. This is actually not the case. An equally good agreement with the recording data can be found in the model in the absence of propagation delay, that is, for $D = 0$. The corresponding wave pattern statistics are shown in *Figure 3—figure supplement 11*, and agree as well as the model with delay with the data for monkey L (*Figures 2 and 3*).

One can wonder how the range $l$ of long-range excitatory connections influences planar wave appearance and properties. This is more easily addressed in the absence of propagation delay ($D = 0$) since, in this case, with other parameters fixed, the operating point of the network (*Figure 1b*) does not change when $l$ is varied, which is not the case when propagation delays are significant. The results of simulations for different $l$ in the model of *Figure 3—figure supplement 11* are shown in *Figure 3—figure supplement 12*.

First, when long-range excitatory connections are absent ($l = 0$), both synchronized states and planar waves are absent and replaced by random states (*Figure 3—figure supplement 12a*). This is

linked to the significant decrease in the correlation of the activities of different modules, shown by the cross-correlation plots in *Figure 3—figure supplement 12a*. This confirms that long-range excitatory connections are required for planar wave existence.

Second, when $I$ is halved (*Figure 3—figure supplement 12b*) as compared to the reference model of *Figure 3* (i.e., $I = 1$ in module distance units or 400µm in real units), both synchronized states and planar waves exist but in reduced proportions as compared to the reference case. Correlatively, the proportion of random states increases. The speed of planar waves is also strongly reduced.

Finally, when $I$ is increased by 50% (*Figure 3—figure supplement 12c*) as compared to the reference model, synchronization between distant modules is increased, the synchronized regime prevails and planar wave occurrence is reduced.

Synchronized and random states are much more prevalent than planar waves and the number of these two types of patterns is comparable ($N = 3906$ synchronized events vs. $N = 4293$ random events in 72 simulations of 10 s duration each). It is interesting that planar waves predominantly arise from synchronized states and moreover that synchronized states tend to persist in the background of planar waves. Namely, of 421 planar waves in SN model simulations that started after a synchronized state, 364 (resp. 57) were followed by a synchronized state (resp. random state), clearly a much greater proportion than if the ending state was drawn at random ($p = 3 \times 10^{-57}$). In contrast, of the 161 planar waves that started from a random state, 92 (69) ended in a random (synchronized) state, a proportion compatible with random draw ($p = 0.22$).

It is difficult to decipher the structure of the inputs associated to planar waves from the existing experimental recordings. The model allows one to more easily address this question for the simulations. To this end, different planar waves episodes in the reference model of *Figure 3* were averaged after aligning their oscillatory phase maps, as described in 'Methods.' In brief, the different events were shifted in time, so that to make their phases at at half-duration coincide, and, where space-dependent maps of planar waves were considered, rotated in space to make the wave direction coincide. This was similarly performed for the experimental recordings. The structure of the mean planar wave obtained by averaging planar wave events propagating along the principal four axis of the grid is shown in *Figure 4*. The average phase (*Figure 4a*) and proxy-LFP (*Figure 4b*) clearly show the planar wave propagation. Similar mean spatio-temporal phase maps are obtained by distinguishing three groups of planar wave velocities before averaging (*Figure 4—figure supplement 1*).

We furthermore aligned planar wave events in time based on wave onset. This temporal alignment procedure can be used to compute the average parameter $\sigma_p$, quantifying synchronization between different module activities ('circular variance of phase,' see *Equation 6* in 'Methods'), associated to the average planar wave around wave onset. As shown in *Figure 4c*, $\sigma_p$ rises just before the planar wave to reach a high synchronization value. Similar phase and LFP profiles are obtained with experimental data (*Figure 4—figure supplement 2a and b*) together with the rise of the synchronization parameter $\sigma_p$ (*Figure 4—figure supplement 2c*).

The average inputs on the modules, which are unavailable in the recordings, can be computed in the model. The global input to the whole network is found to be shifted to negative values before the appearance of a planar wave (*Figure 4d*), effectively reducing the drive to both the excitatory and the inhibitory populations of all modules and pushing the network towards its self-sustained oscillating regime (see *Figure 1f*). When the local inputs are averaged in the same manner, no significant feature emerges. This is presumably due to the large variability of local input patterns that can give rise to a planar wave.

In order to go beyond correlation and test whether a shift of global inputs can help create planar waves, we performed simulations of the reference model in which, in addition to the fluctuating local and global inputs (*Figure 4e*), steps of current were injected in all modules of the network, as depicted in *Figure 4f and g*. The current steps were chosen of a duration of 50 ms and of an amplitude comparable to the global input shift associated to planar waves. The injection of these current steps resulted in a 50% increase of the number of planar wave episodes (*Figure 4f*). This was not simply a result of the perturbation of the network since injection of steps with the opposite sign of the injected current on the contrary strongly reduced the appearance of planar waves (*Figure 4g*).

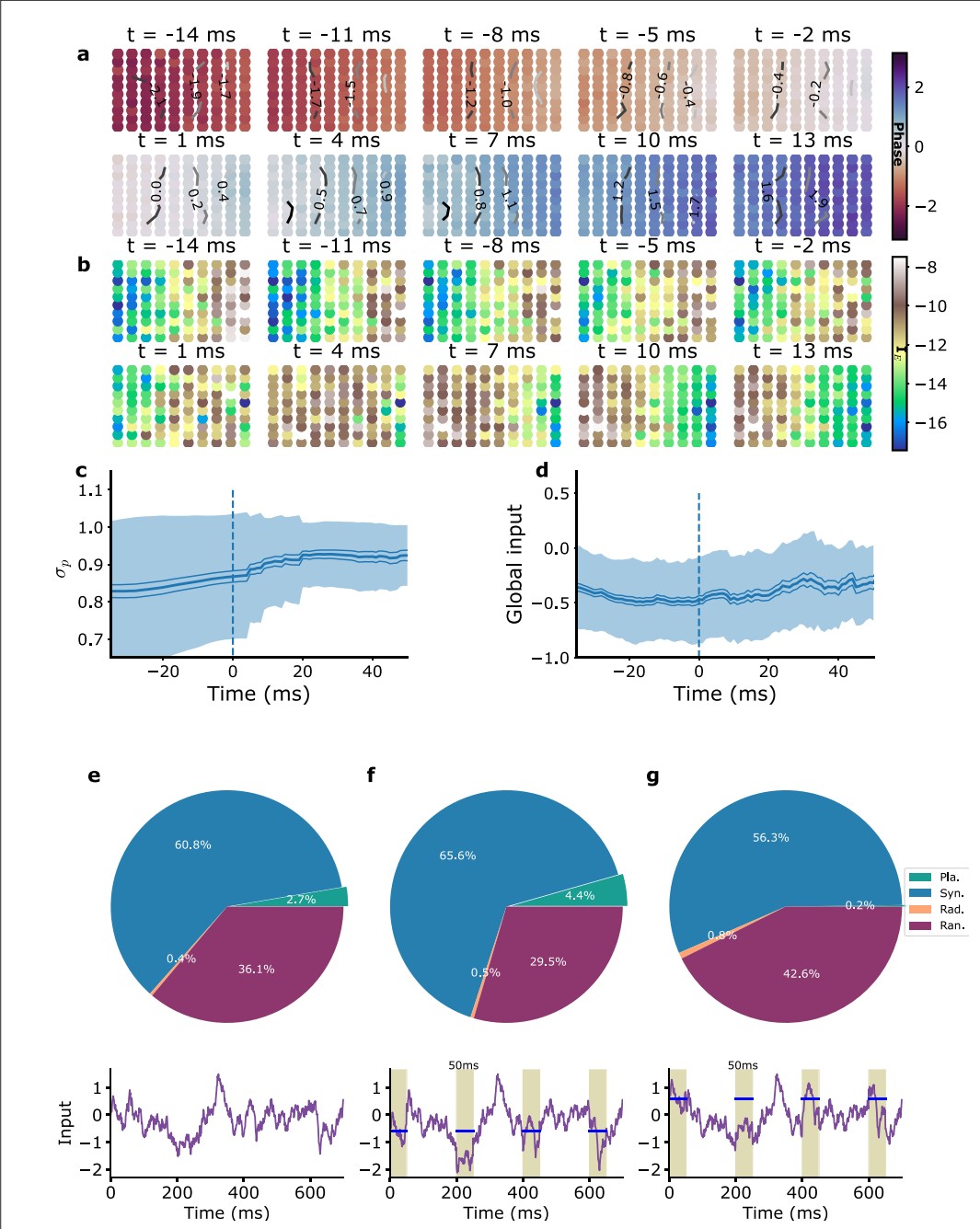

**Figure 4.** Average planar wave characteristics and the role of global inputs. (**a**) Average phase, (**b**) average excitatory current $I_E$ (proxy-LFP), (**c**) average synchronization parameter $\sigma_p$, and (**d**) average global external input. For (**a**, **b**), planar waves were aligned in time by matching their average phases around mid-event, and rotated to make wave directions coincide (see 'Methods'). The time $t = 0$ corresponds to zero average phase. For (**c**, **d**), planar waves were aligned in time based on wave onset (vertical dashed blue line), corresponding to time $t = 0$ in these panels. (**e–g**) Stimulation by global currents modifies planar wave proportion (top). Typical traces of the global inputs including the fluctuating part and the additionally injected current for the three cases are shown in the graphs below (purple lines); periods of injection are shown as shaded areas and injection strength is indicated by the blue bars. (**e**) Without additional stimulation, the mean number of planar wave events per simulation is $N_{pw} = 6.4 \pm 2.3$ with a mean duration of each event $D_{pw} = 42.2 \pm 38.7$ ms ($N = 10$ simulations of 10 s each). (**f**, **g**) Wave proportions with injected current. (**f**) Negative stimulation with an amplitude comparable to that observed in (**d**) produces a greater number of planar wave events $N_{pw} = 9 \pm 2.6$ ($N = 10$, $p = 0.04$ one-tailed Welch's $t$-test), mean duration of planar wave events $D_{pw} = 43.5 \pm 37.8$ ms. (**g**) Positive stimulation decreases the number

*Figure 4 continued on next page*

*Figure 4 continued*

of planar waves events $N_{pw} = 2.8 \pm 1.9$, ($N = 10$, $p = 0.003$ one-tailed Welch's $t$-test), mean duration of planar wave events $D_{pw} = 25.4 \pm 24.7$ ms. The proportion of the different wave types is shown in the three cases.

The online version of this article includes the following figure supplement(s) for figure 4:

**Figure supplement 1.** Average planar wave spatio-temporal phase maps.

**Figure supplement 2.** Average spatio-temporal planar wave maps for monkey L recordings.

## Synaptic connection anisotropy and planar wave propagation direction

Our model network provides a satisfactory description of LFP correlations and wave statistics in the data of *Brochier et al., 2018* when averaged over directions. The long-range synaptic connection probability that we implemented decreases with distance and is independent of orientation. With this choice, we checked that the direction of observed planar waves does not significantly depend on

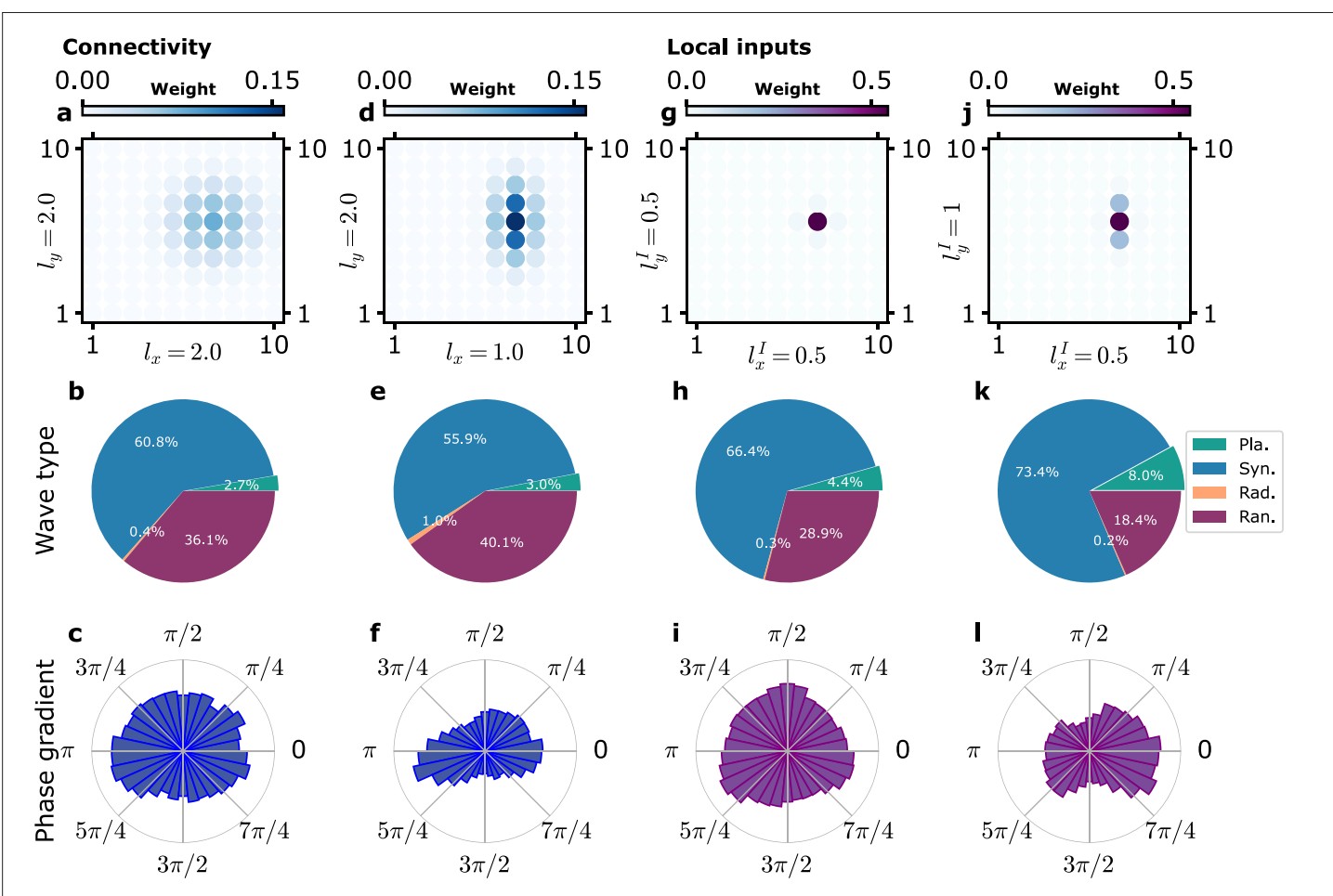

**Figure 5.** Influence of anisotropy of connectivity or local inputs on planar wave propagation direction. (**a–f**) Effect of anisotropic recurrent connectivity. (**a–c**) Model simulations with an isotropic connectivity as in the previous figures. (**d–f**) Model simulations with an anisotropic connectivity. (**a, d**) The function $C(\mathbf{x}, \mathbf{y})$ is shown (with $\mathbf{x}$ arbitrarily chosen at position (7,6)). (**b, e**) Proportion of different wave types. (**c, f**) Distribution of propagation directions of planar waves. In (**f**), the planar waves predominantly propagate along the x-axis, the axis along which $C$ decreases the fastest. (**g–l**) Effect of anisotropy of local inputs. (**g–i**) Model simulations with isotropic local inputs. This is similar to previous figures with the difference that local inputs targets adjacent modules weighted by a Gaussian kernel $G(\mathbf{x}, \mathbf{y})$ (*Equation 22*). (**j–l**) Model simulations with anisotropic local inputs. (**g, j**) The function $G(\mathbf{x}, \mathbf{y})$ is shown (with $\mathbf{x}$ arbitrarily chosen at position (7,6)). (**h, k**) Proportion of different wave types. (**i,l**) Distribution of propagation directions of planar waves. In (**l**), the planar waves predominantly propagate along the x-axis, the axis along which $G$ decreases the fastest.

The online version of this article includes the following figure supplement(s) for figure 5:

**Figure supplement 1.** Model with a generalized description of local inputs.

orientation as shown in *Figure 5a–c* (namely, the anisotropy created by our choice of a rectangular grid is weak). However, it was reported by *Rubino et al., 2006* that waves tend to propagate along preferred axes on the cortical surface with respect to sulcal landmarks and that the orientation of the preferred axes of propagation also vary between different regions, for example, between primary motor cortex and dorsal premotor cortex. For a given preferred axis, propagation in both directions was observed, for example, both rightward and leftward propagating waves were recorded. In order to investigate the possible origin of this observation, we assessed the effect of introducing some anisotropy in our model. We considered two simple possibilities, namely, that intrinsic connectivity was anisotropic or that local inputs target the motor cortex in an anisotropic manner (*Figure 5*). We consider them in turn.

First, in order to analyze the effect of anisotropic connectivity in our model, we simulated a modified version of the model in which the connectivity was of twice longer range in the y-direction than in the x-direction, as illustrated in *Figure 5d*. This did not change much the distribution of the observed wave types (*Figure 5e*). In contrast, *Figure 5f* shows the measured distribution of the directions along which planar waves propagate when connections are anisotropic. The fractions of waves propagating along the x- and y-axis are clearly different. There is a large predominance of waves observed to propagate along the x-axis, namely, along the axis where the connectivity is shorter range. Longer-range connectivity promotes longer-range synchronization along the y-axis and synchronized states. The weaker connectivity along the x-axis allows for the formation of the larger phase gradients along the x-axis and planar waves.

Second, we wished to analyze the effect of a possible anisotropy of the local input spread. In our reference model, local inputs target only one module. In order to be able to introduce anisotropy, we first generalized the local input description. We considered the case where local inputs predominantly target a given module at position $\mathbf{x}$ but also the neighboring ones, at positions $\mathbf{y}$, with smaller weights as described by a kernel function $G(\mathbf{x}, \mathbf{y})$ (*Equation 22* and 'Methods'). As for connectivity, $G$ was taken to be Gaussian. Simulation results for an isotropic $G$ of range $l^{(I)}$ are shown in *Figure 5g–i* and *Figure 5—figure supplement 1*. When $l^{(I)}$ is large enough so that local inputs significantly target neighboring modules, it has a strong synchronizing effect as seen both in the large proportion of synchronized states and the high cross-correlation of the activity of different modules (*Figure 5—figure supplement 1a and b* for $l^{(I)} = 2$ in grid units or 800 μm in real units and *Figure 5—figure supplement 1c and d* for $l^{(I)} = 1$ or 400 μm in real units). For local inputs of narrower footprints (*Figure 5g–i* and *Figure 5—figure supplement 1e and l* for $l^{(I)} = 0.5$ or 200 μm in real units), the results are analogous to the ones obtained with our reference model, as intuitively expected.

With this generalized description of local inputs, we can now consider an anisotropic kernel $G(\mathbf{x}, \mathbf{y})$ (*Equation 22*) so that local input targeting is more spread out in one direction than in the orthogonal one. This is represented in *Figure 5j* for the case that we simulated where local inputs are more spread out in the y-direction ($l_y^{(I)} = 1$ or 400 μm in real units) than in the x-direction ($l_x^{(I)} = 0.5$ or 200 μm in real units). This anisotropic targeting of local inputs produces a clear anisotropy in the distribution of planar wave propagation directions, as shown in *Figure 5l*. Planar waves preferentially propagate along the x-axis where the inputs are confined to a narrower region and gradients in the phase of the oscillatory activity are more easily produced.

Thus, both anisotropy in connectivity and in the targets of local inputs can produce an anisotropy in wave speed direction. However, *Gatter et al., 1978* reported a more spatially extended connectivity in layers 2/3 and 5 of motor cortex in non-human primates in the rostro-caudal direction. *Rubino et al., 2006* found that planar waves predominantly propagated along the same rostro-caudal axis. Given our results, this argues against connectivity anisotropy being the dominant factor determining planar wave speed direction. It remains to be investigated whether local input targets are more spread out along the medio-lateral dimension, as the model suggests.

## Discussion

We have proposed and studied here a simplified model of the motor cortex based on local recurrent connections coupling excitatory and inhibitory neuronal populations together with long-range excitatory connections targeting both populations. This builds on previous works describing beta oscillations in motor cortex as arising from recurrent interactions between excitatory and inhibitory populations (*Pavlides et al., 2015*; *Sherman et al., 2016*; *Chen et al., 2020*). Of note, the model of a

single E-I module of *Sherman et al., 2016* includes two layers of hundred multi-compartment spiking neurons to approximately take into account the layered structure of the cortex and allow a detailed comparison with MEG signals. In a complementary fashion, our model uses a simpler description of an E-I module but goes beyond these previous works by taking into account long-range interactions and analyzing the resulting spatio-temporal patterns.

The well-known oscillatory instability of coupled E-I networks is not qualitatively modified by the presence of long-range connections and the model displays an instability leading to sustained oscillatory behavior. These oscillations occur at beta frequency for adequate synaptic parameters. Comparison with available data has led us to conclude that the motor cortex operates close to this beta oscillatory instability line under strong fluctuating inputs from other areas. This constantly displaces the operating point of the dynamics and leads to waxing and waning of beta amplitude oscillations. A similar proximity of an oscillatory regime has been proposed for the bursts of beta oscillations in the basal ganglia using different models (*Rubchinsky et al., 2012*; *Bahuguna et al., 2020*). Analysis of this oscillatory behavior in time and across electrodes has led us to infer the characteristics of the appropriate external inputs. The observed power enhancement on the low frequency-side of the beta peak requires that their time correlation is not too short on a 10 ms scale while the duration of the observed oscillation bursts requires that it is not too long on the same time scale. In an analogous way, the external input amplitude should be, on the one hand, sufficiently large to significantly modify the LFP characteristics above the background of intrinsic local fluctuations arising from the finite number of neurons sampled by each electrode. On the other hand, it should be small enough to preserve the harmonicity of beta oscillations, as indicated by the absence of a significant secondary 50–60 Hz peak in the LFP spectra. The spatial correlation of beta oscillations at the millimeter distance scale has furthermore led us to suggest that the motor cortex receives inputs that target local areas but also synchronous inputs that target it more globally. Under these conditions, traveling waves of different kinds are observed that resemble those recorded in vivo both in their repartition between the different wave types and their speeds of propagation. Characterization of the inputs onto motor cortex during movement preparation is needed to test these theoretical predictions.

Propagating waves of neural activity have been observed in different contexts (*Ermentrout and Kleinfeld, 2001*; *Muller et al., 2018*). Some are unrelated to neural oscillations such as subthreshold waves (*Bringuier et al., 1999*) or propagation of spiking activity in visual cortex (*Davis et al., 2020*). Others are based on oscillatory activity with proposed mechanisms relying on a gradient of frequencies (*Ermentrout et al., 1998*) or structural sequences of activation (*Muller et al., 2016*). The traveling waves and the mechanism here proposed to underlie them are quite different. They are obviously based on the existence of oscillations since the traveling waves are a reflection of different phases of oscillation at different positions on motor cortex. The model we have developed here includes structural connectivity that tends to synchronize oscillations at different positions, but the stochastic external inputs play an essential role in creating dephasings between different locations. The wave propagation speeds are found to be rather low and distributed in the few tens of cm/s. The mean speed of about 30 cm/s is comparable to the propagation speed along non-myelinated horizontal fibers (*González-Burgos et al., 2000*; *Girard et al., 2001*). However, we have found that it is actually independent of it, as a comparable distribution of traveling speeds is found in a model with no propagation delays. This stands, for instance, in sharp contrast with a recent model of the propagation of spiking activity in visual cortex (*Davis et al., 2021*), which does not involve oscillatory activity but also takes into account strong local fluctuations of neural activity. In the present context, the observed speeds of the oscillatory waves of activity are set by the oscillatory frequency and by the dephasings produced by the external inputs in the recorded area.

We have proposed that external inputs have both local and global components. The existence of a global component appears consistent with the presence of global inhibition in motor cortex during movement preparation (*Greenhouse et al., 2015*). The model 'external inputs' could represent direct inputs from different neural structures, including other cortical areas such as frontal and parietal cortex (*Davare et al., 2011*), and extra-cortical ones like the thalamus (*Hooks et al., 2013*), or a mixture of the two. The known thalamo-cortical connectivity (*Jones, 2001*) makes the thalamus a privileged candidate for the origin of, at least, part of these external inputs. Indeed the described diffused connectivity from calbindin-positive matrix neurons could be the source of our global inputs while core parvalbumin-positive neurons could be the source of our local inputs. This requires further

experiments assessing the origin of synaptic inputs and their influence on waves and wave types. We have moreover suggested that the observed anisotropy of wave propagation is linked to the anisotropy of local input targets, with the specific prediction that traveling waves of oscillatory activity are predominantly observed orthogonal to the axis where they are more spread out. This also needs to be tested in further experiments.

The model suggests that inputs to the motor cortex play an important role in the generation of beta oscillations and of spatio-temporal patterns during movement preparation. The experimental measurement of these inputs and their correlations at different locations thus appears an important research direction. Do synchronized inputs target distant locations of motor cortex? Does a a shift of these global inputs accompany the birth of planar waves as we have found in the model? Perturbation experiments, for instance, by optogenetic means, would also bring important information. Can sustained beta oscillations be induced by mild inhibition of pyramidal cells and interneurons? The model furthermore suggests that such perturbations applied in a transient way would enhance the appearance of planar waves.

Finally, the role of traveling waves during movement preparation remains an important question that needs to be clarified. We have shown that a stochastic description of the inputs to the motor cortex is sufficient to account for the recordings. However, the produced traveling waves are correlated with the particular inputs that the motor cortex receives. Indeed, traveling waves have been shown to carry information about the subsequent movement (*Rubino et al., 2006*). Movement preparation is presently viewed, in a dynamical systems perspective, as taking place in a subspace orthogonal to the dynamics of the movement itself and producing the proper initial condition for it (*Kaufman et al., 2014*; *Wang et al., 2018*; *Zimnik and Churchland, 2021*; *Inagaki et al., 2022*; *Bachschmid-Romano et al., 2022*). Beta oscillations and their associated timescale are however absent from this description. Further work is needed to refine it and include beta oscillations and waves and obtain a more comprehensive description of motor cortex dynamics.

## Methods
### Recording data
We analyzed the data that was made publicly available and described in detail in *Brochier et al., 2018*. We content ourselves here to describe their main features for the convenience of the reader. The data were obtained from two macaque monkeys (L and N) trained to perform one of four movements at a GO signal. In brief, in each trial, the animal was presented for 300 ms with a visual cue that provided partial information on the movement to be performed. It had to wait for 1 s, before the GO signal that also brought the missing information about the rewarded movement. The different phases of a trial are depicted in *Figure 1—figure supplement 1*. The neural activity was recorded during the task with a 10 × 10 square multielectrode Utah array, with neighboring electrodes separated by 400 μm, implanted in the primary motor cortex (M1) or premotor cortex contralateral to the active arm. We make use of two published recording sessions (one for monkey L of 11:49 min/135 correct trials, one for monkey N of 16:43 min/141 correct trials) as fully detailed in *Brochier et al., 2018*.

### Data analysis
An overview of our data analysis protocol is provided in *Figure 3—figure supplement 1*. The different steps are detailed below.

#### Data filtering
The index of each electrode in the data (*Figure 2—figure supplement 2*) was matched to the $(x, y)$-position of the electrode in the data that we use in the following. The few missing electrode signals were replaced by the average of the neighboring electrode signals. The followings steps were applied to the so obtained $N = 100$ electrode signals. The LFP signal of each electrode was band-pass filtered in the 13–30 Hz range using a third-order Butterworth filter (signal.filtfilt function of the Python package scipy). The filtered signal of each electrode was then z-score normalized and Hilbert-transformed, using the scipy Hilbert transform function signal.hilbert to obtain its instantaneous amplitude $A_{xy}(t)$ and phase $\Phi_{xy}(t)$ (*Figure 2—figure supplement 2*). The instantaneous maps $A_{xy}(t)$ and $\Phi_{xy}(t)$ were used for beta burst and wave pattern detection, respectively.

## Beta burst analysis

For each electrode amplitude time trace $A_{xy}(t)$, the percentiles of the amplitude distribution were determined (**Tinkhauser et al., 2017**). The beta burst threshold was set at the 75*th* percentile. A burst onset was defined as a time point at which the analytical amplitude exceeded the threshold and its termination as the earliest following time point at which the amplitude fell below the threshold. The time difference between these two time points was defined as the burst duration. The amplitude of a burst was defined as the average amplitude during the burst duration. The procedure is illustrated in *Figure 2—figure supplement 2*.

## Spatio-temporal pattern analysis and wave classification

The spatio-temporal patterns of oscillatory activity were classified with the help of the phase map $\Phi_{xy}(t)$ following the steps of *Denker et al., 2018* with minor modifications. First, the phase gradient map $\Gamma_{xy}(t)$ and its normalized version, the phase directionality map $\Delta_{xy}(t)$, were obtained. Specifically the phase gradient map was obtained by averaging the phase differences between a point and its existing next and next-nearest neighbors along the x- and y-axis,

$$
\begin{aligned}
\Gamma_{xy}(t) \quad &= \tfrac{1}{N_x} \sum\nolimits_{\text{neighbors with } |dx| \leq 2} \tfrac{\Phi(x+dx,y;t)-\Phi(x,y;t)}{dx} \\
&+ \tfrac{i}{N_y} \sum\nolimits_{\text{neighbors with } |dy| \leq 2} \tfrac{\Phi(x,y+dy;t)-\Phi(x,y;t)}{dy}
\end{aligned}
\tag{2}
$$

where the vector has been written as a complex number and $N_x$, $N_y$ count the actual number of neighbors along x and y, respectively ($N_x + N_y$ may differ from 8 for electrodes near the array boundaries). Normalization of the gradients to unit length serves to produce the 'phase directionality map' (*Denker et al., 2018*),

$$
\Delta_{xy}(t) = |\Gamma_{xy}(t)|^{-1} \Gamma_{xy}(t)
\tag{3}
$$

The alignment of the unit vectors $\Delta_{xy}(t)$ at time $t$ was quantified using the norm $\sigma_g$ of their mean, the 'circular variance of phase directionality' (*Denker et al., 2018*), with

$$
\sigma_g(t) = N^{-1} |\sum\nolimits_{x,y} \Delta_{x,y}(t)| \in [0, 1]
\tag{4}
$$

with $N$ the total number of electrodes.

The data was classified into successive 1 ms spatial 'patterns.' Patterns with well-aligned $\Delta_{xy}(t)$ were classified as planar wave patterns using the criterion $\sigma_g(t) > 0.5$ and provided the first category of the classification.

The remaining patterns were further classified using additional measures. Patterns were next categorized as 'radial waves' (or not) by considering critical points (i.e., maxima, minima or saddle points) of the phase map $\Phi_{xy}(t)$ (*Rule et al., 2018*). A smoothed phase gradient map, the 'gradient coherence map,' was defined following *Denker et al., 2018*,

$$
\Lambda_{xy}(t) = N_{xy}^{-1} \sum\nolimits_{x',y' \in \{-2,-1,0,1,2\}} \Delta_{x+x',y+y'}(t)
\tag{5}
$$

(Note that the sum is taken over existing neighboring electrodes, the number of which is given by $N_{xy} \leq 9$.) Minima, maxima, and saddle points were obtained by locating the sign changes in the local gradient coherence map (*Perry and Chong, 1987*). Maps with exactly one critical point were classified as radial waves, thus providing the second category of our classification (these patterns could be subdivided in further subcategories [*Rule et al., 2018*] but this was not pursued).

The synchronous patterns were finally identified using the 'circular variance of phases' (*Denker et al., 2018*), $\sigma_p$, analogous to the well-known Kuramoto order parameter,

$$
\sigma_p(t) = N^{-1} |\sum\nolimits_{x,y} e^{j\Phi_{x,y}(t)}| \in [0, 1]
\tag{6}
$$

with $N$ the total number of electrodes in the array. Patterns with $\sigma_p(t) > 0.85$, indicating tight synchronization of the oscillations of all electrodes, were classified as 'synchronized,' the third classification category. All remaining patterns were assigned to the 'random' fourth category.

After all successive patterns were classified, a planar, a radial, or a synchronized wave episode was registered when at least six consecutive frame beared such a label. In other words, planar, radial,

or synchronized wave episodes were requested to last for at least 6 ms to be declared as such. All remaining patterns were registered as random waves.

## Alignment and averaging of planar waves

To compute average quantities relating to planar waves like average phase maps or input strength, we aligned events in time in two different ways, either (i) with respect to the phase of the underlying oscillation or (ii) with respect to the onset of a planar wave. For the phase alignment, we determined the average phase across the array at half the duration of any given event, which we denote $\bar{\phi}_{\text{center}}^{(k)}$ for event $k$ in the following. We then converted this phase to a time with respect to a reference frequency $f_{ref}$ according to $t_{\text{center}}^{(k)} = \bar{\phi}_{\text{center}}^{(k)}/(2\pi f_{\text{ref}})$, where $f_{\text{ref}} = \sum_k f_k/K$ is the mean frequency over all $K$ events of a given type, with $f_k$ being given by $\langle d\phi/dt\rangle/(2/\pi)$ averaged over space and time during the event. We then aligned all events in time by centering each event $k$ on time $t_{\text{center}}^{(k)}$ rounded to the closest time bin. With this choice, time $t = 0$ corresponds to an average phase $\bar{\phi} = 0$. For the alignment relative to wave onset, we simply aligned all planar wave events with respect to the first frame classified as planar wave.

To furthermore compute averages over planar waves without loosing spatial information related to wave direction, but not restrict ourselves to the analysis of waves that propagate along a single direction, we considered all wave events that propagate along any of the four principal directions (0, $\pi/2$, $\pi$, $3\pi/2$) of the grid, which are statistically equivalent. We specifically selected waves with a mean gradient vector, averaged over units and wave duration, that points within a $\pm\pi/8$ interval centered on the respective direction. (Note that waves with a mean gradient outside the selected sector are not taken into account.) In a second step, we then rotated units by (0, $3\pi/2$, $\pi$, $\pi/2$) for events of the four principal directions, respectively, such that eventually all gradients pointed to the $\pm\pi/8$ interval centered on 0, corresponding to a leftward traveling wave (opposite to the mean gradient). As all four principal directions are equivalent due to the symmetry of the system, this allowed us to improve the statistics of our measurements without introducing any bias because of the discretized grid.

## Model

The motor cortex is described as a collection of recurrently connected excitatory (E) and inhibitory (I) neuronal populations of linear size $\sim 400\mu m$, comparable to the one of a cortical column. These E-I modules are connected by long-range excitatory connections with a connection probability decaying with the distance between modules. The neural activity of each E-I module is represented at the level of its excitatory and inhibitory neuronal populations in the rate-model framework (*Wilson and Cowan, 1972*; *Dayan and Abbott, 2005*). This eases numerical simulations that would otherwise be extremely demanding for a comparable spatially structured network of a few hundred modules with a few tens of thousand spiking neurons each. Additionally, the rate model description also eases analytical computations. We choose a rate model description with an adaptive time scale (*Ostojic and Brunel, 2011*; *Augustin et al., 2017*). It was shown to quantitatively describe networks of stochastically spiking exponential integrate-and-fire (EIF) neurons, either uncoupled and receiving identical noisy inputs (*Ostojic and Brunel, 2011*), or coupled by recurrent excitation (*Augustin et al., 2017*), as well as sparsely synchronized oscillations of recurrently coupled spiking E-I modules of EIF neurons (*Kulkarni et al., 2020*). The adaptive time scale rate model describes the activity of a population of EIF neurons as

$$\tau(I)\frac{dI}{dt} = -I + I_0 + s(t), \; r(t) = \Phi_\sigma[I(t)] \tag{7}$$

with $\Phi_\sigma[I]$ the f-I curve of an EIF neuron with white noise current input of mean $I$ and noise strength $\sigma$ (see *Kulkarni et al., 2020* for a detailed description). The time scale $\tau(I)$ is referred to as 'adaptive' because it varies with the current $I$. The function $\tau(I)$ is chosen here, as in *Kulkarni et al., 2020* and precisely described there, to best fit the response to oscillatory inputs in a wide frequency range of 1 Hz to 1 kHz, of an EIF neuron subjected also to a white noise current of mean $I$ and strength $\sigma$. The tabulated functions $\Phi_\sigma[I]$ and $\tau(I)$ are plotted in *Figure 1—figure supplement 3*. They are given in *Kulkarni et al., 2020* and are also provided here as *Source data* for the convenience of the reader.

We generalize the rate model description *Equation 7* to a two-dimensional network of E-I modules coupled by long-range excitation,

$$\tau_E(I_E)\frac{dI_E}{dt}(\mathbf{x},t) = -I_E(\mathbf{x},t) + I_E^{ext}(\mathbf{x},t) + I_{EE}^{syn}(\mathbf{x},t) - I_{EI}^{syn}(\mathbf{x},t) \tag{8}$$

$$\tau_I(I_I)\frac{dI_I}{dt}(\mathbf{x},t) = -I_I(\mathbf{x},t) + I_I^{ext}(\mathbf{x},t) + I_{IE}^{syn}(\mathbf{x},t) - I_{II}^{syn}(\mathbf{x},t) \tag{9}$$

where the index denotes the module position on a rectangular grid. The firing rates are related to the currents by

$$r_A(\mathbf{x},t) = \Phi_A[I_A(\mathbf{x},t)] + \sqrt{\frac{\Phi_A[I_A(\mathbf{x},t)]}{N_A}}\xi_A(\mathbf{x},t), \ A \in \{E,I\} \tag{10}$$

The $\xi_A(\mathbf{x},t)$ are independent unit amplitude white noises for each neuronal population in each module $\langle \xi_A(\mathbf{x},t)\xi_B(\mathbf{x}',t')\rangle = \delta_{A,B}\delta_{\mathbf{x},\mathbf{x}'}\delta(t-t')$ and Ito's prescription is used to define *Equation 10* (*Kulkarni et al., 2020*). These stochastic terms account, in the firing rate framework, for the fluctuations due to the finite numbers $N_E$ of excitatory neurons and $N_I$ of inhibitory in each E-I module (*Brunel and Hakim, 1999*). For the numerical computations, both $\Phi_E$ and $\Phi_I$ are taken equal to the function $\Phi_\sigma[I]$ provided in the *Source data* and plotted in *Figure 1—figure supplement 3a*.

The currents $I_E^{ext}(\mathbf{x},t)$ (resp. $I_I^{ext}(\mathbf{x},t)$) represent inputs external to the motor cortex, possibly position and time-dependent, targeting the excitatory (resp. inhibitory) population of the E-I module at position $\mathbf{x}$,

$$I_A^{ext}(t)(\mathbf{x},t) = I_A^{ext,0} + \sigma_A^{ext}\eta(\mathbf{x},t), \ \sigma_A^{ext} = w_A^{ext}\nu_{ext}, \ A \in \{E,I\} \tag{11}$$

With our normalization choice $\nu_{ext}$, has the dimension of a frequency and can be interpreted as the discharge rate amplitude of the external inputs. We model the fluctuations of external inputs as stochastic O-U processes with global and independent components,

$$\begin{aligned}\tau_{ext}\frac{d\eta}{dt}(\mathbf{x},t) &= -\eta(\mathbf{x},t) + \sqrt{\tau_{ext}}\left[\sqrt{1-c}\,\xi(\mathbf{x},t) + \sqrt{c}\,\xi_g(t)\right]\\ \langle\xi(\mathbf{x},t)\xi(\mathbf{x}',t')\rangle &= \delta(t-t')\delta_{\mathbf{x},\mathbf{x}'}, \ \langle\xi_g(t)\xi_g(t')\rangle = \delta(t-t')\end{aligned} \tag{12}$$

The currents $I_{EE}^{syn}(\mathbf{x},t)$, $I_{EI}^{syn}(\mathbf{x},t)$ (resp. $I_{IE}^{syn}(\mathbf{x},t)$, $I_{II}^{syn}(\mathbf{x},t)$) represent the recurrent excitatory and inhibitory inputs on the excitatory (resp. inhibitory) population of the module at position $\mathbf{x}$. The recurrent inputs depend on the firing rates of the neuronal populations of the different motor cortex modules and on the kinetics of the different synapses. Namely,

$$\begin{aligned}I_{AE}^{syn}(\mathbf{x},t) &= w_{AE}\int_{-\infty}^t du\, S_E(t-u)\sum_{\mathbf{y}} C(|\mathbf{x}-\mathbf{y}|)r_E(\mathbf{y},u-D|\mathbf{x}-\mathbf{y}|),\\ I_{AI}^{syn}(\mathbf{x},t) &= w_{AI}\int_{-\infty}^t du\, S_I(t-u)r_I(\mathbf{x},u), \ A \in \{E,I\}\end{aligned} \tag{13}$$

The kinetic kernels $S_E(t)$, $S_I(t)$ include synaptic current rise times, $\tau_r^E,\tau_r^I$, decay times, $\tau_d^E,\tau_d^I$, and latencies, $\tau_l^E,\tau_l^I$,

$$S_A(t) = \frac{\theta(t-\tau_l^A)}{\tau_d^A-\tau_r^A}\left\{\exp[-(t-\tau_l^A)/\tau_d^A] - \exp[(-(t-\tau_l^A)/\tau_r^A)]\right\}, \ A \in \{E,I\} \tag{14}$$

where $\theta(t)$ denotes the Heaviside function, $\theta(t) = 1, t > 0$ and 0 otherwise. The kernels $S_A, A \in \{E,I\}$ are normalized such that:

$$\int dt\, S_A(t) = 1 \tag{15}$$

We have supposed, for simplicity, that the kinetics of the synaptic currents depend on their excitatory or inhibitory character but not on the nature of their post-synaptic targets (e.g., $I_{EI}^{syn}$ and $I_{II}^{syn}$ have the same kinetics). Instead of using the kinetic kernels $S_A(t), A \in \{E,I\}$, one can equivalently compute the synaptic currents by introducing supplementary variables $J_{AE}, J_{AI}$,

$$\begin{aligned}\tau_d^E\frac{dI_{AE}^{syn}}{dt}(\mathbf{x},t) &= -I_{AE}^{syn}(\mathbf{x},t) + J_{AE}(\mathbf{x},t),\\ \tau_r^E\frac{dJ_{AE}}{dt}(\mathbf{x},t) &= -J_{AE}(\mathbf{x},t) + w_{AE}\sum_{\mathbf{y}} C(\mathbf{x}-\mathbf{y})r_E(\mathbf{y},t-\tau_l^E - D|\mathbf{x}-\mathbf{y}|)\end{aligned} \tag{16}$$

$$\begin{aligned}\tau_d^I\frac{dI_{AI}^{syn}}{dt}(\mathbf{x},t) &= -I_{AI}^{syn}(\mathbf{x},t) + J_{AI}(\mathbf{x},t),\\ \tau_r^I\frac{dJ_{AI}}{dt}(\mathbf{x},t) &= -J_{AI}(\mathbf{x},t) + w_{AI}r_I(\mathbf{x},t-\tau_l^I)\end{aligned} \tag{17}$$

In *Equation 13* or *Equation 16*, the probability that an excitatory neuron at position $\mathbf{y}$ targets a neuron at position $\mathbf{x}$ is represented by the function $C(\mathbf{x} - \mathbf{y})$ normalized such that

$$\sum_{\mathbf{x}} C(\mathbf{x}) = 1 \tag{18}$$

The delay $D|\mathbf{x} - \mathbf{y}|$ accounts for the signal propagation time along horizontal non-myelinated fibers in the motor cortex (*González-Burgos et al., 2000*; *Girard et al., 2001*). The mathematical expressions below are written for a general function $C(\mathbf{x})$. For all the numerical computations, except those shown in *Figure 5d–f*, the following isotropic Gaussian function is taken,

$$C(\mathbf{x}) = \tfrac{1}{Z} \exp(-|\mathbf{x}|^2/l^2), \ Z = \sum_{\mathbf{x}} \exp(-|\mathbf{x}|^2/l^2) \tag{19}$$

In *Figure 5d–f*, we consider instead the anisotropic function,

$$C(\mathbf{x}) = \tfrac{1}{Z} \exp(-x^2/l_x^2 - y^2/l_y^2), \ Z = \sum_{\mathbf{x}} \exp(-x^2/l_x^2 - y^2/l_y^2) \tag{20}$$

with $\mathbf{x} = (x, y)$.

In *Figure 5*, an anisotropic distribution of the local input targets is considered. This is modeled by replacing *Equation 12* by

$$
\begin{aligned}
\tau_{ext} \tfrac{d\eta}{dt}(\mathbf{x}, t) &= -\eta(\mathbf{x}, t) + \sqrt{\tau_{ext}} \left[ \sqrt{c} \, \xi_g(t) + \sqrt{1-c} \sum_y G(\mathbf{x}, \mathbf{y}) \xi(\mathbf{y}, t) \right] \\
\langle \xi(\mathbf{x}, t) \xi(\mathbf{x}', t') \rangle &= \delta(t - t') \delta_{\mathbf{x}, \mathbf{x}'}, \ \langle \xi_g(t) \xi_g(t') \rangle = \delta(t - t')
\end{aligned} \tag{21}
$$

The kernel $G$ is taken to be Gaussian,

$$G(\mathbf{x}) = \tfrac{1}{Z} \exp[-(x/l_x^l)^2 - (y/l_y^l)^2], \ Z = \left( \sum_{\mathbf{x}} G^2(\mathbf{x}) \right)^{1/2} \tag{22}$$

where the normalization $Z$ is chosen such as to have the same local variance of the noise as in *Equation 12*. In *Figure 5g–i* and *Figure 5—figure supplement 1*, an isotropic kernel $G$ is taken with $l_x^l = l_y^l = l^l$.

## Theoretical analysis

Our analysis starts by considering the deterministic version of the model described by *Equations 8–17*, that is, the limit $N_E \to \infty$, $N_I \to \infty$, in which the amplitudes of the stochastic terms vanish in *Equation 10*. All module populations are supposed to fire steadily in time at constant rates $r_E^s$ and $r_I^s$. This requires external inputs that are constant in time, that is, with $\sigma_E^{ext} = \sigma_I^{ext} = 0$ (*Equation 11*), and of specific magnitudes that we first determine.

We then assess the stability of this steady state. This provides the bifurcation diagrams of *Figure 1*. When the steady state is stable, we compute the effect of fluctuations arising both from the finite number of neurons in each E-I module and from the time variation of the external inputs. This provides the analytic curves for the power spectra and correlations of currents, shown in *Figure 2*, *Figure 2—figure supplement 1*, and *Figure 2—figure supplement 3*.

### Steady state

We consider first the steady state of the deterministic network in which excitatory populations and inhibitory populations are firing at constant rates $r_E^s$ and $r_I^s$, independently of time and module position, with

$$r_A^s = \Phi_A(I_A^s), \ A \in \{E, I\} \tag{23}$$

In this state, the synaptic currents in the different populations are obviously also independent of time and position. Given the normalization conditions (*Equations 15; 18*), they simply read,

$$I_{AB}^{syn}(\mathbf{x}, t) = I_{AB}^{syn, s} = w_{AB} r_B^s, \ A \in \{E, I\}, B \in \{E, I\} \tag{24}$$

From the model definition (*Equations 8; 9*), these firing rates are produced by constant external inputs of magnitudes,

$$I_A^{ext}(\mathbf{x}, t) = I_A^{ext, s} = I_A^s - w_{AE} \, r_E^s + w_{AI} \, r_I^s, \ A \in \{E, I\} \tag{25}$$

## Bifurcation lines

The stability of the steady firing state can be assessed by imposing the constant external currents (*Equation 25*) and computing the dynamics of small perturbations around the steady state. To this end, we linearize *Equations 8–17* and look for solutions that are oscillatory in space and exponential in time,

$$I_E(\mathbf{x}, t) = I_E^s + \delta I_E(\mathbf{x}, t), \ I_E(\mathbf{x}, t) = \tilde{\delta}I_E(\mathbf{q}, \sigma) \exp(\sigma t + i\mathbf{q} \cdot \mathbf{x}) \tag{26}$$

with similar expansions for the other variables. Substitution in the explicit formulas (*Equation 14*), or in the corresponding differential equations (*Equations 16; 17*), provides the expression of the synaptic current in term of the module activities,

$$\tilde{\delta}I_{AB}^{syn}(\mathbf{q}, \sigma) = w_{AB}C(\mathbf{q}, \sigma)\tilde{S}_B(\sigma)\tilde{\delta}I_B(\mathbf{q}, \sigma) \tag{27}$$

where $\tilde{S}_E(\sigma)$ and $\tilde{S}_I(\sigma)$ are the Laplace transforms of $S_E(t)$ and $S_I(t)$,

$$\tilde{S}_A(\sigma) = \frac{\exp(-\sigma\tau_l^A)}{(1+\sigma\tau_r^A)(1+\sigma\tau_d^A)}, \ A \in \{E, I\} \tag{28}$$

Substitution of *Equation 27* in the linearized *Equations 8; 9* gives

$$\tilde{L}_{EI}(\mathbf{q}, \sigma) \cdot \tilde{\delta}\mathbf{I}(\mathbf{q}, \sigma) = 0 \tag{29}$$

The matrix $\tilde{L}_{EI}(\mathbf{q}, \sigma)$ reads,

$$\tilde{L}_{EI}(\mathbf{q}, \sigma) = \begin{pmatrix} 1 + \sigma\tau_E - \alpha C(\mathbf{q}, \sigma)\tilde{S}_E(\sigma) & w_{EI}\Phi_I'(I_I^s)\tilde{S}_I(\sigma) \\ \\ -w_{IE}\Phi_E'(I_E^s)C(\mathbf{q}, \sigma)\tilde{S}_E(\sigma) & 1 + \sigma\tau_I + \gamma\tilde{S}_I(\sigma) \end{pmatrix} \tag{30}$$

where the function $C(\mathbf{q}, \sigma)$ is the Fourier transform of the coupling function with propagation delays taken into account,

$$C(\mathbf{q}, \sigma) = \sum_{\mathbf{x}} \exp[-i\mathbf{q} \cdot \mathbf{x} - \sigma|\mathbf{x}|D]C(|\mathbf{x}|) \tag{31}$$

In *Equation 30* and in the following, the short-hand notation $\tau_E$ (resp. $\tau_I$) is used for $\tau_E(I_E^s)$ (resp. $\tau_I(I_I^s)$). The existence of a non-trivial solution of *Equation 29* requires that the determinant, $W(\mathbf{q}, \sigma)$, of the matrix $\tilde{L}_{EI}(\mathbf{q}, \sigma)$ vanishes, that is,

$$W(\mathbf{q}, \sigma) = \left[1 - \alpha C(\mathbf{q}, \sigma)\tilde{T}_E(\sigma)\right]\left[1 + \gamma\tilde{T}_I(\sigma)\right] + \beta C(\mathbf{q}, \sigma)\tilde{T}_E(\sigma)\tilde{T}_I(\sigma) = 0 \tag{32}$$

with the functions $\tilde{T}_E(\sigma), \tilde{T}_I(\sigma)$, defined by,

$$\tilde{T}_A(\sigma) = \frac{\tilde{S}_A(\sigma)}{1 + \tau_A\sigma}, \ A \in \{E, I\} \tag{33}$$

The constants $\alpha$ and $\gamma$ respectively measure the gain of monosynaptic recurrent excitation and inhibition while $\beta$ measures the gain of disynaptic recurrent inhibition,

$$\alpha = w_{EE}\Phi_E'(I_E^s), \ \beta = w_{IE}w_{EI}\Phi_E'(I_E^s)\Phi_I'(I_I^s), \ \gamma = w_{II}\Phi_I'(I_I^s) \tag{34}$$

where $\Phi_A'(I)$ denotes the derivative of $\Phi_A$ with respect to $I$.

The oscillatory instability, or 'Hopf bifurcation,' line corresponds to the parameters for which the growth rate is purely imaginary, $\sigma = i\omega$. It is obtained in parametric form, with $\alpha$ and $\beta$ as functions of the frequency $\omega$ (and of the recurrent inhibition $\gamma$) by separating the real and imaginary parts of *Equation 32* and solving the resulting linear equations for $\alpha$ and $\beta$. This gives

$$\begin{aligned} \alpha &= \frac{\text{Im}\left\{ C(\mathbf{q}, i\omega)\left[\tilde{T}_I(i\omega)\tilde{T}_E(i\omega) + \gamma|\tilde{T}_I(i\omega)|^2\tilde{T}_E(i\omega)\right] \right\}}{|C(\mathbf{q}, i\omega)|^2 \, |\tilde{T}_E(i\omega)|^2 \, \text{Im}[\tilde{T}_I(i\omega)]} \\ \beta &= \frac{\text{Im}\left[C(\mathbf{q}, i\omega)\tilde{T}_E(i\omega)\right]|1 + \gamma\tilde{T}_I(i\omega)|^2}{|C(\mathbf{q}, i\omega)|^2 \, |\tilde{T}_E(i\omega)|^2 \, \text{Im}[\tilde{T}_I(i\omega)]} \end{aligned} \tag{35}$$

The instability first appears at long wavelengths, namely, at $\mathbf{q} = 0$ on a large enough lattice. The expressions (35) with $\mathbf{q} = 0$ have been used to draw the oscillatory instability lines and the frequency dependence on parameters shown in *Figure 1b* and *Figure 1—figure supplement 4*.

Besides this oscillatory instability, there is a possible loss of stability towards a high-firing rate state. It is obtained when the growth rate $\sigma$ changes with parameter variation from being real negative to real positive, that is, for parameters such that $W(\mathbf{q}, 0) = 0$. Since at the 'real' instability threshold, the instability growth rate vanishes, the instability line is independent of the synaptic current kinetics. One indeed checks from *Equations 28; 33* that $\tilde{T}_E(0) = 1$. Thus, *Equation 32* gives for the real instability line for a given wavevector $\mathbf{q}$,

$$\beta = [\alpha - \tfrac{1}{C(\mathbf{q},0)}][1 + \gamma], \text{ and } \beta = (\alpha - 1)(1 + \gamma) \text{ when } \mathbf{q} = 0 \tag{36}$$

Since $C(\mathbf{q}, 0) < 1$ when $\mathbf{q}$ does not vanish, the instability appears first at $\mathbf{q} = 0$ with $C(\mathbf{0}, 0) = 1$ when all the modules are in the exact same state.

The expressions (*Equation 35*, *Equation 36*) with $\mathbf{q} = 0$ have been used to draw the diagram of *Figure 1b*. The diagram of *Figure 1f* is similarly obtained by computing the external inputs corresponding to different steady discharges, for fixed synaptic parameters, and assessing the stability of these states from their position with respect to the stability lines (*Equation 35* and *Equation 36*). We have taken the synaptic strength of external inputs such that the network operating point crosses the oscillatory bifurcation line when the external input amplitude varies:

$$w_E^{ext} = w_{EE}, \ w_I^{ext} = 2w_{IE} \tag{37}$$

## Auto and cross-correlations of module activities and power spectrum

We consider the network described by *Equations 8 and 9* with the kinetics of the synaptic currents given by *Equation 14*. We include the stochastic effects arising from the finite number of neurons in each module by using the stochastic description (*Equation 10*) of the instantaneous firing rates of the excitatory and inhibitory module neuronal populations. We also include external input fluctuations as described by *Equations 11; 12*.

We treat these two kinds of stochastic effects as perturbations of the steady dynamics and fully characterize the stochastic dynamics of the network at the linear level.

Linearizing the currents around their stationary values gives

$$I_E(\mathbf{x}, t) = I_E^s + \delta I_E(\mathbf{x}, t), \ I_I(\mathbf{x}, t) = I_I^s + \delta I_I(\mathbf{x}, t) \tag{38}$$

Translation invariance leads us to search for $\delta I_E$ and $\delta I_I$ in Fourier space:

$$\delta I_E(\mathbf{x}, t) = \int_{-\pi}^{+\pi} \int_{-\pi}^{+\pi} \tfrac{dq_x}{2\pi} \tfrac{dq_y}{2\pi} \int_{-\infty}^{+\infty} \tfrac{d\omega}{2\pi} \tilde{\delta} I_E(\mathbf{q}, \omega) \exp[i(\mathbf{q} \cdot \mathbf{x} + \omega t)] \tag{39}$$

with an analogous expansion for $\delta I_I$. The linearized *Equations 8; 9* then read, with a vectorial notation

$$\tilde{L}_{EI}(\mathbf{q}, i\omega) \cdot \tilde{\delta}\mathbf{I}(\mathbf{q}, i\omega) = \mathbf{F}(\mathbf{q}, i\omega) \tag{40}$$

where the $2 \times 2$ matrix $\tilde{L}_{EI}(\mathbf{q}, i\omega)$ is given in *Equation 30*. The two components of the stochastic forcing term $\mathbf{F}(\mathbf{q}, i\omega)$ read

$$
\begin{aligned}
F_A(\mathbf{q}, i\omega) \ &= w_{AE} C(\mathbf{q}, i\omega) \tilde{S}_E(i\omega) \sqrt{\tfrac{r_E^s}{N_E}} \, \tilde{\xi}_E(\mathbf{q}, \omega) \\
&- w_{AI} \tilde{S}_I(i\omega) \sqrt{\tfrac{r_I^s}{N_I}} \, \tilde{\xi}_I(\mathbf{q}, \omega) + \sigma_A^{ext} \tilde{\eta}(\mathbf{q}, \omega), \ A \in \{E, I\}
\end{aligned}
\tag{41}
$$

where $\sigma_A^{ext}$ is given by *Equations 11; 37*. Solution of the linear system (40) provides the expression of the Fourier components of the currents. For the excitatory current, one obtains

$$\tilde{\delta} I_E(\mathbf{q}, \omega) = \tfrac{1}{(1 + i\omega\tau_E)} \tfrac{V_E(\mathbf{q}, i\omega)}{W(\mathbf{q}, i\omega)} \tag{42}$$

with $W(\mathbf{q}, i\omega)$ defined in *Equation 32* and $V_E(\mathbf{q}, i\omega)$ given by,

$$V_E(\mathbf{q}, i\omega) = F_E(\mathbf{q}, i\omega) \left[1 + \gamma \tilde{T}_I(i\omega)\right] - F_I(\mathbf{q}, i\omega) w_{EI} \Phi_I' \tilde{T}_I(i\omega) \tag{43}$$

The Fourier components of the input fluctuations read

$$\tilde{\eta}(\mathbf{q}, \omega) = \frac{\sqrt{\tau_{ext}}}{1+i\omega\tau_{ext}} \left[ \sqrt{1-c}\, \tilde{\xi}(\mathbf{q}, \omega) + \sqrt{c}\,(2\pi)^2\delta^2(\mathbf{q})\tilde{\xi}_g(\omega) \right] \tag{44}$$

with the white noise averages,

$$\langle \tilde{\xi}(\mathbf{q},\omega)\tilde{\xi}^*(\mathbf{q}',\omega')\rangle = (2\pi)^3\delta(\omega-\omega')\delta^2(\mathbf{q}-\mathbf{q}'), \quad \langle \tilde{\xi}_g(\omega)\tilde{\xi}_g^*(\omega')\rangle = 2\pi\delta(\omega-\omega') \tag{45}$$

The short-hand notation $\delta^2(\mathbf{q})$ has been used for the two-dimensional $\delta$-function $\delta(q_x)\delta(q_y)$. Similarly, one has for the finite size noise averages:

$$\langle \tilde{\xi}_E(\mathbf{q},\omega)\tilde{\xi}_E^*(\mathbf{q}',\omega')\rangle = \langle \tilde{\xi}_I(\mathbf{q},\omega)\tilde{\xi}_I^*(\mathbf{q}',\omega')\rangle = (2\pi)^3\delta(\omega-\omega')\delta^2(\mathbf{q}-\mathbf{q}') \tag{46}$$

The current–current correlation function is obtained by averaging the product of the currents (*Equation 42*) over the noise with the help of *Equations 45; 46*. One obtains,

$$\langle \tilde{\delta I}_E(\mathbf{q},\omega)\tilde{\delta I}_E^*(\mathbf{q}',\omega')\rangle = 2\pi\delta(\omega-\omega')\{(2\pi)^2\delta^2(\mathbf{q}-\mathbf{q}')S_{EE}^N(\mathbf{q},\omega)$$
$$+[(1-c)(2\pi)^2\delta^2(\mathbf{q}-\mathbf{q}')+c(2\pi)^4\delta^2(\mathbf{q})\delta^2(\mathbf{q}')]S_{EE}^{ext}(\mathbf{q},\omega)\} \tag{47}$$

with

$$S_{EE}^{ext}(\mathbf{q},\omega) = \frac{\tau_{ext}\left|\sigma_E^{ext}+(\gamma\sigma_E^{ext}-\sigma_I^{ext}w_{EI}\Phi_I')\tilde{T}_I(i\omega)\right|^2}{[1+(\omega\tau_{ext})^2][1+(\omega\tau_E)^2]|W(\mathbf{q},i\omega)|^2} \tag{48}$$

and

$$S_{EE}^N(\mathbf{q},\omega) = \frac{1}{|W(\mathbf{q},i\omega)|^2}\left\{ \frac{r_E^s}{N_E}w_{EE}^2\left|1+(\gamma-\frac{\beta}{\alpha})\tilde{T}_I(i\omega)\right|^2\left|C(\mathbf{q},i\omega)\tilde{T}_E(i\omega)\right|^2 \right.$$
$$\left. + \frac{r_I^s}{N_I}\frac{w_{EI}^2|\tilde{S}_I(i\omega)|^2}{1+(\omega\tau_E)^2} \right\} \tag{49}$$

This provides the expression of the current–current correlation in real space:

$$\langle \tilde{\delta I}_E(\mathbf{x},t)\tilde{\delta I}_E(\mathbf{x}',t')\rangle = \int_{-\infty}^{+\infty}\frac{d\omega}{2\pi}\exp[i\omega(t-t')]\Big\{ c\,S_{EE}^{ext}(\mathbf{0},\omega)$$
$$+ \int_{-\pi}^{+\pi}\int_{-\pi}^{+\pi}\frac{dq_x}{2\pi}\frac{dq_y}{2\pi}\left[(1-c)S_{EE}^{ext}(\mathbf{q},\omega)+S_{EE}^N(\mathbf{q},\omega)\right]\exp[i\mathbf{q}\cdot(\mathbf{x}-\mathbf{x}')]\Big\} \tag{50}$$

One can check that the cross-correlation is a real function, as it should, since $S_{EE}^{ext}$ and $S_{EE}^N$ are related to their complex conjugates by $S_{EE}^{ext*}(\mathbf{q},\omega) = S_{EE}^{ext}(-\mathbf{q},-\omega)$ and $S_{EE}^{N*}(\mathbf{q},\omega) = S_{EE}^N(-\mathbf{q},-\omega)$, a symmetry inherited from the function $C(\mathbf{q},\omega)$ (*Equation 31*).

Taking coincident points (i.e., $\mathbf{x}=\mathbf{x}'$) in the current–current correlation (*Equation 50*) gives the $I_E$ current auto-correlation for a local module in the network. Remembering that the auto-correlation is the Fourier transform of the spectrum, this provides the spectrum $S(\omega)$ of a local module excitatory current time series:

$$S(\omega) = c\,S_{EE}^{ext}(\mathbf{0},\omega) + \int_{-\pi}^{+\pi}\int_{-\pi}^{+\pi}\frac{dq_x}{2\pi}\frac{dq_y}{2\pi}\left[(1-c)S_{EE}^{ext}(\mathbf{q},\omega)+S_{EE}^N(\mathbf{q},\omega)\right] \tag{51}$$

We take the local module excitatory current as a proxy for the LFP. *Equations 50; 51* have been used to draw the theoretical lines for current–current correlations and power spectra in *Figure 2*, *Figure 2—figure supplement 1* and *Figure 2—figure supplement 3*.

## Simulations

The mathematical model defined in section 'Model' was numerically simulated. We used a 28 × 28 grid of E-I modules. Two external layers of with E-I populations at their fixed points were added as boundary conditions. Measurements were only performed in the center square 10 × 10 grid to minimize boundary effects. A sketch of the grid is shown in *Figure 1—figure supplement 2*. Model distributions and averages in all figures were obtained by performing 20 independent network simulations of 10 s simulated time each.

The reference parameters for all simulations are provided in *Table 1*.

Simulations were performed with a custom C program using a first-order Euler–Maruyama integration method, with a time step $dt = 0.01$ ms. Python programs were used for data analysis and to draw the figures. These computer codes are available at https://github.com/LKANG777/Beta-Oscillation (copy archived at *Kang, 2023*). All simulations were performed on ECNU computer clusters.

## Acknowledgements

We are very grateful to M Denker for extensive discussions about the analysis in *Denker et al., 2018* and for allowing us to precisely compare it with our own. It is a pleasure to thank Nicolas Brunel and David Hansel for thought-provoking comments, Nicolas Brunel and Ludovica Bachschmid-Romano for sharing their unpublished work on the motor cortex and Betsy Herbert for her work and discussions during an internship on a related project. We also wish to acknowledge exchanges with Thomas Brochier and Alexa Riehle, and thank Nicholas Hatsopoulos for pointing out to us the data on anisotropic connectivity in *Gatter et al., 1978*.

## Additional information

### Funding

| Funder | Grant reference number | Author |
| --- | --- | --- |
| China Scholarship Council | Graduate Student Fellowship | Ling Kang |

The funders had no role in study design, data collection and interpretation, or the decision to submit the work for publication.

### Author contributions

Ling Kang, Software, Formal analysis, Funding acquisition, Investigation, Visualization, Writing – original draft; Jonas Ranft, Conceptualization, Formal analysis, Supervision, Investigation, Visualization, Writing - review and editing; Vincent Hakim, Conceptualization, Formal analysis, Supervision, Funding acquisition, Writing – original draft, Writing - review and editing

### Author ORCIDs

Ling Kang http://orcid.org/0000-0001-6532-3773
Jonas Ranft http://orcid.org/0000-0002-7843-7443
Vincent Hakim http://orcid.org/0000-0001-7505-8192

### Decision letter and Author response

Decision letter https://doi.org/10.7554/eLife.81446.sa1
Author response https://doi.org/10.7554/eLife.81446.sa2

## Additional files

### Supplementary files
• MDAR checklist

### Data availability

The source codes for this manuscript are available on GitHub at https://github.com/LKANG777/Beta-Oscillation (copy archived at *Kang, 2023*).

The following previously published dataset was used:

| Author(s) | Year | Dataset title | Dataset URL | Database and Identifier |
|---|---|---|---|---|
| Brochier T, Zehl L, Hao Y, Duret M, Sprenger J, Denker M, Grün S, Riehle A | 2017 | Massively parallel recordings in macaque motor cortex during an instructed delayed reach-to-grasp task | https://doi.org/10.12751/g-node.f83565 | G-Node, 10.12751/g-node.f83565 |

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
