## [Editor Report]

This article makes a valuable contribution to the field. Here the authors have developed a convincing model to characterize the generation of motor cortex β oscillations. Using the model, the authors are able to recapitulate several properties observed experimentally. Given the long-standing interest in motor cortical β oscillations and current interest in traveling waves, this article will be of significant interest to the neuroscience community.

---

## [Decision Letter]

**Decision letter after peer review:**

Thank you for submitting your article "Βeta oscillations and waves in motor cortex can be accounted for by the interplay of spatially-structured connectivity and fluctuating inputs" for consideration by *eLife*. Your article has been reviewed by 3 peer reviewers, and the evaluation has been overseen by a Reviewing Editor and Laura Colgin as the Senior Editor. The following individuals involved in review of your submission have agreed to reveal their identity: Thomas Knopfel (Reviewer #1); Nicholas G Hatsopoulos (Reviewer #3).

Essential revisions:

(1) When introducing the multi-electrode arrays as well as when describing the model layout, spatial dimensions should be stated upfront.

(2) The abbreviations for the wave types (Pla., Syn., Rad., Ran. ) need to be expanded and introduced appropriately.

(3) Regarding the "external inputs": the design of the model would allow for interpreting the external inputs as inputs from other cortical areas or a mixed cortical-extracortical input. Similarly, the global and nonglobal inputs may have different mixtures of cortical and extracortical components. This should be stated explicitly.

(4) The authors repetitively state a relatively slow propagation speed of 0.3 m/s along unmyelinated horizontal axons (with no supporting reference). This value is correct for horizontal cerebellar parallel fibers (which are 0.1 μm in diameter) but to my knowledge not for the thicker horizontal long-range axons in the cerebral cortex. This should be discussed/modified.

(5) Could you show the spatiotemporal structure of an average of traveling waves (average across same type and perhaps a limited range of feature parameters)?

(6) Anatomical data on connectivity (type, spatial decay and anisotropies) should be derived from literature and these empirical data should be used to consider the plausibility of the corresponding model assumptions.

(7) Can you derive tractable tests to evaluate the proposed model against alternative published models and can you propose new tractable experimental research questions based on the insights from your study?

(8) It would be helpful to have an intuitive explanation on what are the parameters affecting the β frequency of oscillations in the main text.

(9) In the Methods, the authors state that they consider the local module of excitatory current as a proxy for the LFP; this should be stated in the main text too, for clarity.

(10) The authors mention that reference (Davis et al., 2021) "also relies on external inputs". Isn't that model without an external drive, only relying on self-sustained dynamics?

(11) On lines 325-327, the authors state that the SN model leads to a proportion of planar wave states of a few percent. Figures 3l shows that the proportion of planar waves is actually quite large, >60%.

*Reviewer #2 (Recommendations for the authors):*

Related work

l 178-179: "These networks that operate close to the bifurcation line thus appear as promising candidates to describe waxing-and-waning of β oscillations seen in recording data"

The authors may be interested to know that similar analysis/observations have been made in the E-I subcircuit of the basal ganglia (STN-GPe) loop by two studies maybe indicating a common mechanism of β bursts across cortical and subcortical circuits.

- Rubchinsky LL, Park C, Worth RM. Intermittent neural synchronization in Parkinson's disease. Nonlinear Dynamics. 2012; p. 329-346. pmid:22582010 –

- Bahuguna J, Sahasranamam A, Kumar A (2020) Uncoupling the roles of firing rates and spike bursts in shaping the STN-GPe β band oscillations. PLOS Computational Biology 16(3): e1007748. https://doi.org/10.1371/journal.pcbi.1007748

---

## [Author Response]

Essential revisions:(1) When introducing the multi-electrode arrays as well as when describing the model layout, spatial dimensions should be stated upfront.

The dimensions of the multi-electrode array and the simulation grid are now described at the end of the Introduction (lines 103-104) and in the first paragraph of the Results section (lines 119-120). We also refer there to Figure 1-Figure supp.2 depicting the simulation grid.

(2) The abbreviations for the wave types (Pla., Syn., Rad., Ran. ) need to be expanded and introduced appropriately.

We have defined the abbreviations in the text (lines 325-326) and in the caption of Figure 3 where they appear for the first time.

(3) Regarding the "external inputs": the design of the model would allow for interpreting the external inputs as inputs from other cortical areas or a mixed cortical-extracortical input. Similarly, the global and nonglobal inputs may have different mixtures of cortical and extracortical components. This should be stated explicitly.

We have now stated this explicitly in the Results first section (lines 245-246) and in the discussion (lines 610).

(4) The authors repetitively state a relatively slow propagation speed of 0.3 m/s along unmyelinated horizontal axons (with no supporting reference). This value is correct for horizontal cerebellar parallel fibers (which are 0.1 μm in diameter) but to my knowledge not for the thicker horizontal long-range axons in the cerebral cortex. This should be discussed/modified.

We apologize for having omitted to give references supporting our choice of conduction velocity. The conduction velocity along horizontal unmyelinated axons have been measured in different works with results in the range 0.10.6m/s. Most of the works we know in the cortex of monkeys are performed in visual cortex. We actually based our choice on Girard, Hup´e & Bullier, J Neurophysiol (2001) who find a median conduction velocity 0*.*3 m/s along horizontal unmyelinated axon at physiological temperature. In Guillermo Gonzalez-Burgos et al., Cerebral Cortex (2000) the mean horizontal axonal conduction velocity was measured in monkey prefrontal cortex to be 0*.*14 m/s at room temperature (20^◦^ − 24^◦^C). It seems reasonable to attribute the difference, as the authors do, to the difference in experimental temperature.

Using voltage sensitive dyes, Grinvald et al., J Neurosci (1994) found the speed of propagation along lateral connections (“apparent conduction velocity” which may underestimate the true conduction velocity) to be in the range of 0*.*09−0*.*25 m/s. Finally, in the rat visual cortex, Lohmann & Rorig, J Comp Neurol (1994) report a mean conduction velocity of 0*.*28 m/s and in rat motor cortex, Aronadiou & Keller, J Physiol (1993) report a conduction velocity of 0*.*1 m/s. However, it is unclear to us whether data obtained in rats could be directly applied to monkey motor cortex. Given these data, it seems that a choice of conduction velocity of 0*.*3 m/s is reasonable and not an underestimate. We have now cited (lines 126-127) the references pertaining to monkey cortex which seem the most relevant for our study. Nonetheless, we are aware of the possibility that the conduction velocity could be larger in monkey motor cortex. That is the reason why we investigated the other extreme case in which propagation delays are actually negligible. This leads us to conclude that the precise value of the velocity does not impact our conclusions.

(5) Could you show the spatiotemporal structure of an average of traveling waves (average across same type and perhaps a limited range of feature parameters)?

In order to address this suggestion and related questions by the referees, we have aligned different planar wave events using their computed spatiotemporal phase map, both for simulations and monkey data. With the help of this alignment, we have computed the average over events of the LFP (or its proxy in simulations) and of the synchronization parameter *σ_p_*. Additionally for the simulations, we have computed the averages of the local inputs as well as of the global inputs, which are not available in the recording data. The structure of the planar traveling waves clearly appear in the average phase and (proxy) LFP maps. We see that the synchronization parameter rises concomitantly with the appearance of the planar waves both in simulations and in experiments. Remarkably, in simulations, this corresponds to an excursion of the global input away from its mean value, which persists during the wave episode. We did not observe any significant average features in the local inputs when averaged over events (not shown), presumably because there is a large variability of local inputs that can contribute to the appearance of a planar wave. The results of this analysis are described in the new section “Properties of planar waves an characterization of their inputs” and shown in the new Figure 4 and Figure 4-Figure supp. 1, and in Figure 4-Figure supp. 2 for the experimental recordings.

In order to try and go beyond this purely correlative observations, we have also performed simulations in which transient steps of global inputs have been injected, with a magnitude similar to the observed excursion. This indeed results in a significant increase of planar wave episodes, as shown in Figure 4e-g.

(6) Anatomical data on connectivity (type, spatial decay and anisotropies) should be derived from literature and these empirical data should be used to consider the plausibility of the corresponding model assumptions.

Precise anatomical data on connectivity in monkey motor cortex is not abundant. We base our choices (lines 122-125) on Gatter and Powell, Brain (1978), Huntley and Jones, J Neurophys (1991), Hao et al., Front Neural Circ (2016) and on the data of Capaday et al., J Neurophys (2009) on cat motor cortex. We thank referee 3 for pointing out the data on anisotropic connectivity in Gatter and Powell, Brain (1978) that we now take into account and discuss in the revised section “Synaptic connection anisotropy and planar wave propagation direction”.

(7) Can you derive tractable tests to evaluate the proposed model against alternative published models and can you propose new tractable experimental research questions based on the insights from your study?

This is of course an important question. We are not aware of other models specifically aimed at describing waves during episodes of β oscillations in the motor cortex. More generally, several models have been developed to describe the propagation of neural activity but among those, few are targeted to traveling waves of oscillatory activity, namely oscillations at different spatial location with a dephasing that depends on space. The ones we know and cite are based on a gradient of excitability/natural oscillation frequency, like for instance the study of Ermentrout et al., J Neurophys (1998) aimed at describing traveling oscillatory activity in the *Limax* olfactory lobe, or on inputs targeting different locations with different structural delays, as the model developed by Muller et al., *eLife* (2016) to describe rotating waves during sleep spindles. That the oscillatory waves of β oscillations in motor cortex are linked to such structural features appears unlikely to us since propagation appears to take place in different directions and particularly along the two opposite directions of the same axis. This led us to propose the model in the manuscript. Since the model relies on inputs from other cortical or subcortical areas, measuring these would offer important tests of the model. We assume that there are global synchronized inputs to the motor cortex. Showing that this is indeed the case would be a first important test of the model. We now show in the revised manuscript (new Figure 4) that shortly before the appearance of a planar wave, there is a shift in the strength of the global inputs. Experimental observation of this effect would bring further support to our model.

Optogenetic perturbations could provide further tests. Our model predicts that changing the strength of the global inputs would allow one to put the motor cortex in a regime of sustained oscillations at β frequency. This could in principle be tested by retroviral injections and optogenetic techniques. As described in point (5) above, we also show in the revised manuscript that creating a diminution of the global external inputs similar to the spontaneous decrease observed in our simulations at the onset of planar waves, increases the probability of appearance of a planar wave (new Figure 4). This could also be experimentally attempted. Finally, on the anatomical side the model suggests that local inputs preferentially target the cortex in the medio-lateral direction.

We have added a paragraph to the Discussion section (lines 619-627) to discuss these experimental research questions suggested by our study.

(8) It would be helpful to have an intuitive explanation on what are the parameters affecting the β frequency of oscillations in the main text.

We have added a paragraph (lines 152-162) to explain that increasing recurrent excitation slows down the oscillation, that increasing recurrent inhibition among inhibitory interneurones increases the oscillation frequency and to discuss the influence of the kinetics of the synaptic current. We have added Figure 1-Figure supp. 4 to show how these parameters affect the oscillation frequency and change.

(9) In the Methods, the authors state that they consider the local module of excitatory current as a proxy for the LFP; this should be stated in the main text too, for clarity.

This has now been stated in the main text (lines 230-232). We now refer to the model signal as the proxy-LFP to avoid any misunderstanding.

(10) The authors mention that reference (Davis et al., 2021) "also relies on external inputs". Isn't that model without an external drive, only relying on self-sustained dynamics?

This has been corrected. We are sorry for the mis-formulation. We meant to write “also take into account strong local fluctuations of neural activity”.

(11) On lines 325-327, the authors state that the SN model leads to a proportion of planar wave states of a few percent. Figures 3l shows that the proportion of planar waves is actually quite large, >60%.

In Figure 3l, the proportion of planar corresponds to the green sector, it is equal to 2.7% which we indeed describe as a few percent in the text. The 60 % blue sector corresponds to the proportion of synchronized states. The color code is indicated in the insert. The abbreviations (Pla, Syn,..) have now been defined in the caption as well as in the main text (lines 325-326).